# Multiplicity of Research Programs in the Biological Systematics: A Case for Scientific Pluralism

**Igor Y. Pavlinov**

Research Zoological Museum, Lomonosov Moscow State University, 125009 Moscow, Russia;
igor_pavlinov@zmmu.msu.ru

**Abstract:** Biological diversity (BD) explored by biological systematics is a complex yet organized natural phenomenon and can be partitioned into several aspects, defined naturally with reference to various causal factors structuring biota. These BD aspects are studied by particular research programs based on specific taxonomic theories (TTs). They provide, in total, a framework for comprehending the structure of biological systematics and its multi-aspect relations to other fields of biology. General principles of individualizing BD aspects and construing TTs as quasi-axiomatics are briefly considered. It is stressed that each TT is characterized by a specific combination of interrelated ontological and epistemological premises most adequate to the BD aspect a TT deals with. The following contemporary research programs in systematics are recognized and characterized in brief: phenetic, rational (with several subprograms), numerical, typological (with several subprograms), biosystematic, biomorphic, phylogenetic (with several subprograms), and evo-devo. From a scientific pluralism perspective, all of these research programs, if related to naturally defined particular BD aspects, are of the same biological and scientific significance. They elaborate "locally" natural classifications that can be united by a generalized faceted classification.

**Keywords:** research programs; scientific pluralism; taxonomic theory; taxonomic pluralisms; typology; phylogenetics; biosystematics; numerical taxonomy; biomorphics; evo-devo

---

## 1. Introduction: Monism vs. Pluralism in Biological Systematics

The dilemma of scientific monism vs. scientific pluralism arose simultaneously with the beginning of modern science development. It can be represented, in a brief and simplified form, as follows [1,2]. In the first case, it is presumed that, both in science in general and in any science branch, there might and should be the only one actually scientific approach providing the only one "right" theory or concept most adequately describing the cognizable world. In the second case, it is presumed that any natural phenomenon, be it entire nature or any of her manifestations, is too complex to be embraced by only one approach and respective theory/concept. Correspondingly, such a "patched" phenomenon might and should be described by several partial theories/concepts, with each capturing its particular manifestation. Therefore, it is their combination that provides an integrated representation of the phenomenon in question. Respectively, the science appeared to be "patched" by research programs, with each developing a particular approach most relevant to a certain manifestation of the respective phenomenon being perceived [3]. Acknowledging such partitioning of both the natural phenomena and scientific activities is of fundamental importance for understanding how various disciplinary fields are individualized and interrelated.

In the life sciences, biological systematics (or systematic biology) plays a fundamental role in such partitioning of biological matter. It studies taxonomic diversity of living beings, as one of the manifestations of the total biological diversity (a.k.a. biodiversity, in its widely adopted scientific sense), which was defined by specific relations between organisms expressed in and revealed by analysis of the

diversity of their proper ("inner") features. Other manifestations of biodiversity, defined by relations of organisms with their environments, are studied by other biological disciplines (biogeography, synecology, sociobiology, etc.). The main function of systematics is to reveal and describe the structure of taxonomic diversity and, thus, to shape, by and large, the subject areas for many other biological disciplines. Due to this, it attracts the attention of both philosophers and biologists exploring the foundations of biology and some of its key concepts such as evolution, hierarchy, species, etc. [4–11].

Taxonomic diversity itself is a multifaceted natural phenomenon because of the multiplicity of relations between organisms shaping it and organismal features expressing these relations. Therefore, systematics develops a variety of approaches for studying the various manifestations of taxonomic diversity. The most important are formalized as taxonomic theories and developed into research programs peculiar to this discipline. They differ in their ontological and epistemological foundations, in their principles of defining objects and tasks, in the methods of exploration, and in the modes of representation of the structure of biodiversity by classifications. These programs change with the development of systematics and biology, depending largely on changes of general scientific-philosophical contexts. In its turn, the changes of the programs dominating at one or another stage in the history of systematics have a significant impact on understanding how biodiversity is structured and, accordingly, what the structure of the entire subject area of biology is.

Since systematics deals with the same natural phenomenon (the above-mentioned taxonomic diversity), a principal question arises, namely, whether it has to follow the only universal research program or whether there can be several programs equally viable. These two positions, which are of a rather philosophical nature, are known as taxonomic monism and taxonomic pluralism, respectively, and they have been actively discussed over the last several decades [2,8,11].

Taxonomic monism presumes that taxonomic diversity should be considered within a unified conceptual approach and described by a single "right" classification, be it either the Natural system of earlier classics (Linnaeus), the general-purpose classification of phenetics (Gilmour), or the phylogenetic classification of phylogenetics (Hennig). The most devoted adherents of each of these approaches, being monists, believe that it is their theory that says the "final word" in biological systematics and should be accepted as such by the entire taxonomic (and eventually biological) community.

On the contrary, taxonomic pluralism acknowledges that taxonomic diversity is multifold and that all its manifestations are of more or less equal biological meaning and scientific (cognitive) status. Respectively, neither taxonomic theories and research programs dealing with such particular manifestations may pretend to gain a privileged position in biological systematics. Each is significant by their careful examination of particular manifestations of taxonomic diversity. For instance, phylogenetic program structures taxonomic diversity historically, while typological program does that structurally, and biomorphic program uncovers complex morpho-ecological aspects of taxonomic diversity.

The first attempts to recognize explicitly and to discuss the research programs in systematics, as well as to evaluate their scientific status, were undertaken in the 1960s–1970s. There appeared to be acknowledged but three principal "systematic philosophies" most vividly discussed at that time, namely, phenetics, cladistics, and evolutionary taxonomy [5,6,12–14]. However, this "three philosophies" viewpoint did not take into consideration or drastically reduce the significance of other "philosophies" that were not so actively discussed then (typology, biosystematics, ecomorphological approach, etc.). Therefore, such an oversimplification provided a very distorted representation of the theoretical foundations of biological systematics, including diversity of the research programs actually operating in it, their historical and scientific-philosophical roots, their mutual interactions and influences, and their contributions toward the development of both systematics itself and biology overall.

In this article, a brief overview of the principal research programs in biological systematics is provided, guided by their scientific and philosophical foundations, regardless of their general acceptance [11]. One of the main tasks is to show why and how these research programs in biological systematics are developed, and why it is normal for the latter to be pluralistic in this respect. Therefore, attention is paid toward ontological and epistemological prerequisites for elaborating taxonomic

theories and for developing the research programs implementing them. The next key task is to show the real diversity of these programs, contrary to a received viewpoint reducing it to a minimal level. All these tasks explain a pretty extensive list of references, even though it only includes the most important issues.

In accordance with the above tasks, the present overview begins with the illumination of some basic ideas concerning the philosophical foundations of biological systematics, as they are understood by the author. This includes also consideration of some general principles of construing taxonomic theory.

## 2. Some Basic Elements of the Systematic Philosophy

Any general philosophy of science deals primarily with the justification of theoretical knowledge in science. Thus, the "philosophy of systematics" actually presumes analysis of the possible ways to develop the theoretical foundations of biological systematics as a whole.

If systematics is considered a scientifically sound discipline, it is to develop its own taxonomic theory (TT). However, despite a significant number of books, in which principles or foundations of systematics are considered, no sufficiently well-substantiated TT has been elaborated so far. Moreover, hardly any satisfactory understanding seems to exist among taxonomists as to what kind of theory it should or could be. There were only a few earlier attempts to consider some basic premises and principles of what might be called the beginnings of TT, but they were too formal to be biologically sound [7,15,16]. The main reason for such a very deplorable situation is that the substantiations of particular classification approaches have been predominated previously, instead of developing the TT in its general understanding, which would cover the entirety of biological systematics. Such a theory should deal with the multiplicity of both the manifestations of biodiversity and the ways to study it.

Any serious consideration of this important issue would first require an ascertainment of how theories of various levels of generality can be built in different scientific disciplines. However, this would lead us away from the main topic of the present paper, especially when taking into consideration the diversity of the viewpoints on this matter. Therefore, without going deep into this issue, it seems to be enough, for the purposes of this article, to offer the following general declarations concerning just the taxonomic theory [11,17]. The latter is defined as a conceptual system containing generalized theoretical knowledge about what and how biological systematics explores. The answers to the question "what?" make up the ontological part of the theory: they fix the essential characteristics of the object of systematic research. The answers to the question "how?" constitute the epistemological part of the theory: they fix the basic principles of the study of this object. Together, these answers constitute a conceptual framework for defining a TT.

It follows that the main purpose of any TT is framing the theoretical context in which systematic research is conducted and concrete classifications are developed. As such, this theory serves as a general basis for the formation and functioning of any research program in biological systematics. Development of such a theory is the most important and an ultimate aim of taxonomy as a theoretical part of biological systematics.

### 2.1. One Umgebung—Many Umwelts

One of the first tasks of TT, in its most general sense, is defining an object (or a subject area) of the entirety of biological systematics. Many serious problems arise related not even to the theory itself, but rather to the philosophy of systematics—not in the above Hull's [13], but in a more general sense, concerning its ontological foundations. They deal with illumination of what this biological discipline and its various sections investigate.

All natural sciences, by an initial condition, are aimed ultimately at comprehending and describing nature in its entirety. However, since nature is so global and diverse in all its manifestations, while the human cognitive possibilities are so limited, it is fundamentally impossible to embrace it as a whole with a single gaze and to reach a complete knowledge of nature expressed by a single exhaustive

generalization. Instead of a "global" comprehension of nature, it is its "local" manifestations (particular aspects, fragments, etc.) that are actually being explored.

According to the contemporary conceptualism, with Quine's concept of ontological relativity constituting one of its cornerstones [18,19], such fragmentation of nature is shaped by a knowing subject (be it either a person or a scientific community) depending on the latter's cognitive tasks. In this connection, distinguishing the two main levels in reality suggested by the Baltic German zoopsychologist von Uexküll [20,21] seems to be very attractive and thought-provoking. According to his idea, the whole of nature as such (Umgebung) and its actually perceived manifestations (Umwelts) are to be distinguished in a cognitive situation dealt with by natural science. Although Uexküll himself meant selective "biological" perceptions of the environments by particular organisms, his rather metaphoric concept may be widened to include human cognitive activity, which is no less selective with respect to nature being perceived [22].

Thus, from a conceptualist perspective, each particular cognitive Umwelt is fixed by a knowing subject in the course of the latter's cognitive activity to shape a kind of conceptual reality not existing out of this cognitive and perceptual activity. As such, it represents a particular cognizable reality, and multiplicity of these Umwelts implies multiplicity of the approaches dealing with them. Therefore, any cognitive activity begins with preliminarily outlining a particular Umwelt and continues with exploring and describing its properties. In the case of biological systematics, taxonomic diversity can be treated as its Umgebung, while its various manifestations are its various respective Umwelts.

Since systematics deals with taxonomic diversity as a natural phenomenon, its particular Umwelts are to be distinguished as naturally as possible. The best way to ensure this seems to be to do so by indicating the supposed objective causes (initial, proximal, formal, material, etc.) structuring the biota, and, thus, generating various manifestations of diversity of organisms. It is presumed that a cognitive Umwelt thus outlined corresponds to a certain biological phenomenon, which makes its exploration biologically meaningful. On the contrary, an Umwelt fixed by a formal ontology, i.e., without reference to certain natural causes, is apparently devoid of biological meaning.

Different cognitive Umwelts corresponding to particular manifestations of taxonomic diversity are shaped by certain concepts, each defining the properties most significant for individualizing the respective Umwelts. For example, this may be diversity of archetypes or biomorphs, or the hierarchy of monophyletic groups, etc. The phylogeny that generates the latter hierarchy can be interpreted classically in its wholeness (according to Haeckel or Simpson) or reduced to cladogenesis (according to Hennig). On the contrary, one can discard all prior considerations of the structure and causes of taxonomic diversity and deal with separate physically perceived organisms. This also requires specific background knowledge, though, with very poor ontology greatly trimmed by means of the "Occam's razor".

As is evident from the preceding, any transition from the whole Umgebung to one or another particular Umwelt is based on a reduction operation, as far as any isolation of a part from its whole means reduction of the latter. In systematics, such a reduction begins with "cutting" something called biodiversity from the entirety of nature. At the next reduction step, taxonomic diversity is singled out from the entirety of biodiversity, with the latter's other manifestations (ecological, biogeographic, biosocial, etc.) being discarded. Then taxonomic diversity undergoes further decomposition by distinguishing its own aspects, e.g., phylogenetic (emphasis on kinship), typological (emphasis on structural plans), ecogenetic (emphasis on diversity of populations), etc. Furthermore, within phylogenetically defined diversity, its cladogenetic and anagenetic aspects can be distinguished and analyzed separately.

Thus, the above transition from overall Umgebung to particular Umwelts can be represented as a kind of reduction cascade at different steps of which particular exploratory tasks of different levels of generality are successively formulated and solved. All of this is accomplished by a knowing subject (as specified above) with the help of various epistemic tools. It is this subject that decides what is significant and what is not for distinguishing particular Umwelts and extracting them from

the Umgebung. Evidently, such a stepwise reduction leads to an unavoidable sequential loss of some part of the objective content of the entire Umgebung at each reduction step. Therefore, an ontology defined by this content is the richest at the very beginning of the reduction cascade and most poor at its end. Accordingly, in the same direction, an accumulated effect of a subjective "input" into particular conceptually construed Umwelts increases. This issue seems to be of prime importance for understanding of "naturalness" (in the above sense) of particular manifestations of taxonomic diversity and, respectively, biological meaningfulness of different taxonomic theories and research programs studying them (see the next section).

It is also important to stress that such a reduction is potentially multiplied at every step of the cascade. This is because taxonomic diversity as a complexly organized natural phenomenon can be represented by several more simple cognitive models (in their general epistemic sense from Reference [23]). Such a multiplicity of the Umwelts, recognized at each step of the reduction, is an ontological prerequisite of an increase of taxonomic plurality descending from top to bottom of the reduction cascade.

## 2.2. Taxonomic Theory as a Quasi-Axiomatics

Every natural science theory includes some elements of axiomatics. This means that it contains more or less clearly formulated statements about the subject being studied (analogues of axioms) and the principles of its research (analogues of inference rules). Considered from a philosophical standpoint, the former constitute ontology, and the latter constitute epistemology. Such a construction of a taxonomic theory (TT) by using at least some elements of the axiomatic method is advantageous in that it allows us to formulate its basic statements more explicitly, and, thus, to structure the theory itself.

Attempts of this kind were undertaken repeatedly. Some were aimed at developing universal mostly formal systems [7,15,16], while others dealt with the foundations of particular research programs (phenetic, phylogenetic, etc., i.e., [24–27]). Here, the author's position is presented very briefly to show why and how taxonomic theories underlying research programs in biological systematics can be structured and justified [11,17]. The main concern of this section is not to suggest a version of the TT but rather to consider some general principles of its development.

It must be emphasized first that a TT is to be developed as a quasi-axiomatics. This means that, unlike formal axiomatic systems of mathematics and logic, its basic conceptual constructs are initially introduced as biologically sound. This is provided by direct reference to a certain objective (real) manifestation of biodiversity (i.e., typological or phylogenetic pattern), which establishes desirable correspondence between an Umwelt (naturally defined, see previous section) and respective set of quasi-axioms, which makes the latter biologically meaningful. It is quite important to stress that the same meaning can be ascribed to classifications based on these quasi-axioms, while formal axioms with no reference to an Umwelt make respective classifications also formal (biologically meaningless).

Usually, in various systematic textbooks, all such theoretical premises are called principles without distinguishing between their different cognitive functions. However, as seen from the example above, in the framework of the axiomatic approach, it is necessary to divide them into two main categories, namely, quasi-axioms and inference rules. The former have an ontological status and outline an Umwelt under study, while the latter have an epistemological status and deal with the principles of investigation of this Umwelt. Only these inference rules seem to deserve being termed "taxonomic principles".

Though an axiomatic method of construing any theory presumes strict and unambiguous definitions, this requirement cannot be followed literally in the case of natural science disciplines including biological systematics. Its implementation is limited by the principle of an inverse relationship between the rigor and the meaningfulness of the concept definition [28]. The more strictly a concept is defined, the less likely there is something in nature to which it may correspond [29]. Therefore, any definition of an Unmwelt, claimed to be biologically meaningful, is deemed to be imprecise semantically and should be formulated by taking into consideration some conditions of the fuzzy logic (as it is defined by Kosko [30]). The latter means, among others, that such "fuzzy" definitions

seem to entail an unfeasibility of their strict and unequivocal applications in studying the structure of biodiversity. The biological concepts that come to mind as most pertinent to this issue are those of taxonomic rank [8,11,31], homology [32–35], and, of course, species [8–10,36–40]. The impossibility of their strict and unambiguous definitions leads to the fuzziness of these concepts, which is reflected by a plurality of their particular meanings.

It is to be stressed especially that quasi-axioms and inference rules within a TT do not work separately, but conjointly in a single package. In general, this condition is formalized by the principle of onto-epistemic correspondence, which means that the basic (for the given TT) statements relating to ontology and epistemology should be meaningfully compatible with each other [11,17]. For example, if an Umwelt is defined phylogenetically, then the principles specify how a classification should be developed to reflect the phylogenetic pattern only.

It follows from the above stepwise reduction cascade (see Section 2.1) that the sequential reduction of the overall Umgebung to a certain set of Umwelts results in the generation of respective particular quasi-axiomatics of different levels of generality. It is presumed that each set of quasi-axioms outlines a particular Umwelts of a certain level of generality. On this basis, a hierarchy of TTs allocated to these levels can be consequently construed. Thus, taxonomic theory considered in its most general sense is a rather complex multilayer construct. It consists of several levels of theoretical generalizations with each solving specific tasks allocated to a respective level of a reduction cascade. Theoretical provisions of the highest level of generality constitute the general taxonomic theory (GTT), while those belonging to the lower levels are particular taxonomic theories (PTTs).

In this hierarchical conceptual pyramid, the GTT plays the role of a framework concept in relation to various PTTs and can be considered a taxonomic meta theory (i.e., theory of theories) for them. Within such a pyramid, particular PTTs arise as different details of the GTT statements. The main task of the latter is to outline correctly (including being biologically sound) the cognitive situation for the entirety of biological systematics, including its basic ontological and epistemological components. Thus, GTT can be imagined as a set of interconnected general judgments about the general properties of taxonomic diversity (ontology) and the general principles of its study (epistemology). This theory is intended not to elaborate concrete classifications, but rather to formulate (as prescriptions and restrictions) the grounds for possible ways to formulate and solve the exploratory tasks that systematics deals with. Thus, it is the GTT that can more than justifiably be claimed as a systematic philosophy.

There are two principal modes of construing the GTT. One of them refers primarily to ontological quasi-axioms from which certain principles are deduced, while another accentuates epistemological reference rules (principles) equally applied to any natural phenomena. Thus, taxonomic pluralism is observed even at the most general level of the theoretical basis of biological systematics.

The ontology-based GTT specifies first how a particular Umwelt is to be outlined. For instance, it specifies whether causes of the diversity of organisms should be indicated or not, and if indicated, which particular ones—historical in phylogenetics, structural in typology, or functional in biomorphic, etc. If the evolutionary process is referred to as the main cause of taxonomic diversity, it can be interpreted as an adaptatiogenesis or as a more "formal" cladogenesis. On this basis, it is then specified (quasi-axiomatized) which particular relations between organisms are taken into account—only kinship, only similarity, or some combinations thereof. Based on these basic assumptions, certain taxonomic principles are developed. Some of them entail homologization, character weighting, similarity assessment, etc., while others deal with inferring particular phylogenetic schemes, and the next deal with elaborating phylogenetic classifications based on these schemes. The same general design is true for any other biologically meaningful quasi-axiomatics, be it typological, biomorphic, or otherwise.

On the other hand, the epistemology-based GTT presumes that any Umwelts, however defined, are not specific with respect to their properties (ontology), so the main task is to elaborate certain universal principles of their analysis (epistemology) independent of particular ontologies. The latter means that systematic research should be subordinated to certain fairly formalized universal reference

rules. Particular implementations of such accentuation are PTTs in which logical or computational procedures are set as being of primary importance (see below).

From a formal standpoint, any properly construed quasi-axiomatic systems relevant to the task systematics deals with have an equal cognitive potential. However, as stressed above, as far as systematics is a biological discipline exploring biodiversity as a real phenomenon, evaluating the research potentials of all possible TTs is to be based on their biological meaningfulness. The latter means, first, that their basic quasi-axioms should refer to certain natural biological phenomena, and, second, the respective Umwelts representing the latter should be as least reductionist as possible. Therefore, ontology-based TTs seem to be more significant as compared to epistemology-based TTs, and, among the former, those with a rich ontology are more significant.

As seen, the outlined quasi-axiomatic method of developing the theoretical foundations of biological systematics makes it rather easy to understand the whole structure of both the GTT (which is still an imaginary, rather than a well-established, construct) and the reasons of plurality of PTTs detailing the latter.

## 3. An Overview of the Research Programs in Systematics

Research programs in biological systematics appear and function as a means of implementation of particular taxonomic theories. The latter develop not by themselves, but in a certain philosophical–scientific context, with one way or another responding to the challenges that systematics faces at one or another stage of conceptual development of natural science.

The subsequent sections provide a review of the research programs in biological systematics that have developed over the 20th and at the beginning of the 21st centuries. Some of them basically continued the ideas formed in the 19th century (typological, phylogenetic, etc.) while others emerged de novo in this period (phenetic, numerical, evo-devo, etc.). It is to be stressed that this review is based on preceding considerations of the principles of construing and evaluating the corresponding taxonomic theories. Accordingly, in characterizing the latter, the most focus is put primarily onto their philosophical (ontological and epistemological) foundations, regardless of the popularity they enjoy among biologists at various stages of the development of biological systematics. For this, the order of presentation of these programs corresponds basically to a certain scale of the richness of their ontologies. The below account opens the most reductionist programs (phenetic, rational, numerical) and closes with the biologically soundest programs (typological, biomorphic, phylogenetic, evo-devo).

### 3.1. The Phenetic Program

This program develops and formalizes an old idea of empirical knowledge and, as such, goes back to folk systematics (see References [11,41,42] on the latter). The beginning of its persistent formation in scientific systematics was laid down by the works of the anti-scholastics of the second half of the 18th century. Usually, the French naturalist M. Adanson's approach is mentioned in this connection—to the extent that the founders of modern numerical phenetics used to call it "neo-Adansonian" [24,25]. However, such identification was shown to be incorrect [5,43], as the Adansonian methodology actually foreruns one of the numerical cladistic approaches [44]. The genuine phenetic concept was expressed at that time by the German naturalist J. Blumenbach. He stated in his "Handbuch der Naturgeschichte" that the "animals that are similar in 19 structures and differ only in the twentieth should be grouped together" (see Reference [11]).

In the 20th century, the phenetic program in its rather strict sense was substantiated by a classification theory based on the positivist philosophy of science, as its early ideologists stated explicitly [24,25,45]. It is closely related to the numerical program (see Section 3.3), so they are often considered conjointly. However, this is not correct. The phenetic theory deals with what is studied (ontology), while the numeric one deals with how that "what" is studied (epistemology).

According to this philosophy, in a cognitive situation of phenetic systematics, the background knowledge is minimized in order to exclude its "metaphysical" content (such a reference to evolution).

Respectively, a phenetically defined Umwelt is simply a set of observable physical bodies with their characters, i.e., organisms. At the same time, the subjective influence is also excluded as much as possible. All operations on those "physical bodies"—their description, comparison, etc.—should be depersonalized and reduced to some elementary automated actions. Phenetic classification is developed as purely empirical (in a philosophical sense). It should be nothing more than a generalization of the observed facts, which makes it independent of any biological theories.

In developing phenetic classifications, the only basis for grouping organisms is their mutual similarity as such, which is assumed to be both objective and theory-neutral. However, both these presumptions are not true, as any similar relation is established by a knowing subject depending on the conceptually framed exploratory tasks [46–49]. Thus, the phenetic program seems to lack its initial "as-if empiric" philosophical background. This similarity is evaluated across the totality of the unit (elementary) characters used in the comparisons without any preliminary assessment of their significance ("weight"). One of the most serious restrictions on the choice of characters is that they should describe organisms themselves (morphology, physiology, genetics, etc.), but not their relations to the environment (ecology, etc.) [24,25]. Such a "weighting" ascends the essentialism of earlier (scholastic) taxonomic theory [11]. The resulting taxa in phenetic classifications are designated as phenons [46–49]. Ontological interpretation of both them and their ranks is nominalistic.

The main purpose of phenetic classification is quite pragmatic. It should not reflect some mysterious "naturalness", but instead should be "useful". The usefulness of a classification depends on its informativeness, i.e., on the volume of information about the diversity of the organisms contained within it. Maximizing the information content in the classification is achieved by increasing the number of characters used for its elaboration. Roughly speaking, the more characters, the better. This condition is substantiated by the positivist principle of total evidence coupled with an ad hoc hypothesis of character non-specificity and the mathematical principle of convergence. It is assumed that classifications, if starting from different initial sets of characters, should converge asymptotically with the maximum possible increase of the number of characters [24,25].

The purely technical nature of phenetic classifications means that they are not evaluated from the point of view of their naturalness in its traditional meaning. Instead, it is replaced by certain operational criteria of the classification informativeness assessment (Gilmour-naturalness). The most informative classifications that can be used to solve many different tasks are termed general purpose ones. With reference to the above principle of convergence, the potential attainability of a single stable reference system as an ideal of phenetic systematics is supposed. Along with it, various special purpose classifications can and should be elaborated to solve certain particular research and applied tasks. There can be a lot of them and they can be very different in their information content, but all of them are subordinate with respect to the general-purpose classification. Thus, the phenetic program is monistic with respect to the general-purpose classification and pluralistic with respect to the special-purpose ones.

The phenetic program, supplied with the numerical methods, was most popular in the 1960s. At present, it has been supplanted by the phylogenetic program in its cladistic and molecular versions (see Section 3.7). In this regard, it is important to keep in mind that the latter borrowed some important points of the phenetic theory. Thus, in molecular phylogenetics, an idea of the reduction of an organism to a set of automatically identified unit characters appeared to be perfected: these are nucleotide base pairs in DNA and RNA sequences. In cladistics, clear elements of the phenetic theory are introduced by the above mentioned principle of total evidence according to which, roughly speaking, "the more characters, the better" [50–52].

From the point of view of the philosophy of science that focuses on modern conceptualism (in the above sense of Reference [18]), the main problem of the positivism based phenetic program is that a strictly empirical knowledge, devoid of any theoretical basis referring to a certain natural ontology, is impossible in the natural science [18,53]. This key standpoint means that both the phenetic theory

itself and the program implementing it dropped out of the framework of contemporary science with its dominating post-positivist philosophy.

### 3.2. The Rational Program

Rationality, understood in its general sense, constitutes the very basis of the science distinguishing it from other forms of the cognitive activity. Among various versions of scientific rationality [54,55], a deductive one is the most relevant for substantiation of the rational program in systematics, as it is understood by the author [11,56]. It is based on acknowledging the paramount importance of such syllogistics in which particular judgments are deduced from general ones. It is presumed that the truth of the former entails the truth of the latter. The general aim of such an approach in biological systematics is to develop a kind of rational classification, which may be classified as "all elements of which are derived on the basis of some general principles, certain theory" [57]. It is clear that formulation of taxonomic theory as a kind of quasi-axiomatics (see Section 2.2) fits completely the conditions of such rationality.

Initial judgments, with which formation of the rational program in systematics began, are two-fold. Some of them are related to the object being researched, i.e., to ontology: this can be termed as an ontic rationality. Others relate to the principles of research, i.e., to epistemology: accordingly, they constitute an epistemic rationality. Thus, this rational program is divided into two subprograms, which are called onto-rational and episto-rational ones. They are similar in the deductive (with reference to the concepts of higher levels of generality) substantiation of particular judgments but differ in the content of the general ones [11,56].

One of the first versions of the onto-rational subprogram in systematics was proposed by the Swiss botanist A.-P. de Candolle (actually, he was probably the first to coin the term "rational classification" in biology) at the beginning of the 19th century, who based his theory on the principles of symmetry borrowed from crystallography [58,59]. In a more general form, the idea of rational systematics was formulated 100 years later by the German natural philosopher H. Driesch [60]. He called for uncovering some general law of orderliness of diversity of biological forms that would be analogous to the natural laws of orderliness of chemical elements in physical chemistry or geometric figures in geometry. A rational classification based on such a law is presumed to become a powerful heuristic that allows certain predictions about still unknown forms. Thus, it is reasonable to call the onto-rational systematics nomothetic [61]. It reveals the general laws of taxonomic diversity instead of presenting the latter as a list of taxa with their diagnostic characters. It is evident that the taxa recognized within the onto-rational taxonomic theory are interpreted realistically as natural kinds in the sense of W. Quine and others [62,63].

Implementation of one of Driesch's ideas led to an aspiration to elaborate parametric classifications of living forms analogous to the above periodic system of elements in chemistry [64,65]. The latter means that such classifications should not be construed on a strictly hierarchical (vertical) basis, but rather horizontal relations between biological forms being most important. The main problem in this case is that the biological forms are much more complex than the chemical elements. Therefore, it is difficult to recognize a key parameter by which their periodic system could be arranged. Such a parameter is usually suggested to define the organismal complexity, but it does not lend itself to a universal, satisfactory enough definition, which allows for the development of a single scale of complexity [66,67]. Thus, the principal idea of this version of the onto-rational program in systematics seems hardly resolvable in general. However, some of its applications are of certain interest as they may uncover some important properties in the ordered structure of biodiversity.

There is another partial taxonomic theory called rational by its creators [68–70], which fits the conditions of onto-rationality. In this case, rationality presumes deducing classifications from the orderliness of the diversity of ontogenetic patterns. In this case, this version is considered a part of the epigenetic typological subprogram (see Section 3.4).

The episto-rational subprogram focuses on certain general inference rules governing the particular exploratory procedures in systematics. Such a standpoint presumes that it is these rules that are primary, and all taxonomic principles and methods are to be inferred from them, while ontological considerations are minimized. In the classical systematics of the 16th to 18th centuries, this general conception led to a fundamental monistic idea of the natural method, a proper application of which would provide the natural classification. According to M. Adanson [71], this method "should be universal or overall, i.e., there should be no exception for it". This general idea was first implemented by scholastic systematics of the 16th to the 18th centuries in a form of the universal genus-species scheme [11]. Post-scholastic systematics rejected this particular scheme, but the very idea was retained and led eventually to the development of two research subprograms. In one of these, the logical argumentation is taken as the basis, while another accentuates on mathematically based judgments.

An idea of strict subordination of taxonomic principles of biological systematics to some general logic was expressed in recent times by several authors [15,16,57,72]. As a matter of fact, it explicates a rather old idea that any classification is but a logical procedure. Some fragmentary attempts to implement it were based on application of the requirements of the formal axiomatic systems (see Section 2.2). A more recent and quite developed general solution is offered by classiology [73]. Ideologists of the "logical systematics" presume that any classification theory derived from some "general logic" is applicable to any phenomena (natural, social, etc.) studied by any classifying sciences, regardless of their natural ontology. However, no one pure logical taxonomic theory for biological systematics has been proposed so far, and no biological classifications of such a kind are known. Thus, the main idea of a would-be logical research program in systematics still remains only a kind of "declaration of intent".

In considering the productivity of such a theory for systematics, of prime importance is the fundamental issue whether its basic idea has any relevance to the tasks this biological discipline deals with. As a matter of fact, systematics asks a question about objective (real) biodiversity and tries to answer it in a sound manner. However, every logical system is merely a specific tool designed to ensure the logical consistency of derivative particular statements with respect to the more general ones, all within a particular formal axiomatic system. Such a tool asserts the logical truth of the conclusions with respect to that system, but it does not say anything about their natural truth with respect to the reality being studied because it does not consider this respect at all [74]. Thus, the program in question does not seem to expect either the very question or an answer as to how "logical" classifications may relate to the reality and, if they relate, how to ascertain this.

If the above declaration is followed literally, it is to be taken into consideration that any appeal to some general logic looks very naïve, as there exist many formal logical systems [74,75], including some (binary, probabilistic, fuzzy, modal, etc.) that are relevant to systematics. Thus, the next key question arises—now it is about the basis for a choice of particular logical systems for deducing particular taxonomic theories applicable for elaborating biologically meaningful classifications.

As far as such a question is concerned, one of the possible answers is provided by the above principle of onto-epistemic correspondence (see Section 2.2). According to the modern conceptualist standpoint [18] underlying this principle, it is ontology that drives epistemology (including logical inference rules), and not vice versa. In other words, the principles of elaboration of biologically meaningful classifications are to be inferred from background assumptions about properties of taxonomic diversity rather than from any pure logic [11,17,76]. Thus, it becomes clear from this standpoint that any kind of a pure logical research program in biological systematics seems to be unfeasible.

A part of the episto-rational subprogram is the numerical one based primarily on the mathematical foundations of taxonomic principles and methods. Because of its significant influence on the development of biological systematics in the 20th century, it likely deserves the status of a research program of its own, so it is considered in the following subsection.

### 3.3. The Numerical Program

As noted, this is actually one of the versions of a wider episto-rational subprogram, though deserving a status of a distinct program. Its source lies in the natural philosophical idea that the "book of nature is written in the language of mathematics" announced by the Italian physicist G. Galilei in the 17th century [77]. Based on this, the Prussian philosopher I. Kant at the end of the 18th century expressed one of the key ideas of modern physicalism: "in any special doctrine of nature, there can be only as much proper science as there is mathematics therein" [78].

The 19th century English naturalist H. Strickland should likely be considered one of the forerunners of the numerical program. Based on the then rather popular so called "taxonomic map" metaphor [79], he likened the similarity between groups of organisms to the distance between territories on a geographical map: the longer the distance, the less the similarity [80]. At the beginning of the 20th century, the Russian biologist E. Smirnov put the key idea of this program this way: you need to "establish those rules and laws that determine the relative position of the phenomena studied. The expression of these laws in the form of mathematical formulas is the highest goal that systematization strives for" [81]. Smirnov called such a taxonomy "exact". In the future, its supporters called it "numerical" and then, quite straightforwardly, "mathematical". The general design of the numerical program in taxonomy was framed mostly in the 1960s–1970s [11,82]. Based on the above Kant's aphorism, adherents of this program consider it the only one deserving the title of scientific in the physicalist sense of the latter and deem it as the most significant scientific revolution in contemporary systematics.

The main content of this program can be formulated in two general principles. First, relations (similarity, kinship, etc.) between organisms and sets thereof can and should be measured quantitatively. Second, the structure of relations thus quantified can and should be transformed into a classification by means of quantitative methods. Implementation of this program is provided by many methods developed to solve various particular classificatory tasks within the general numerical idea.

It is clear that the numerical program, based mainly on epistemology, does not have any subject domain of its own. It concerns the issues related to ontology as far as an Umwelt to be studied is to be adapted to the needs of quantitative methods by means of its significant reduction. For instance, the organisms are reduced to a matrix of an uncorrelated formalized character. Thus, any function of this program is limited to serve as an analytical supplement to research programs based on ontological considerations—only those that consider such a reduction is acceptable.

Depending on particular biologically meaningful tasks solved using quantitative methods, the program under consideration is divided into two main subprograms: viz. numerical phenetic and numerical phyletic ones.

Numerical phenetics [24,25,83,84] provides quantitative methods for implementation of the phenetic and, partly, the bio-systematic programs (see Sections 3.1 and 3.6 on them). In this case, a classification procedure is based on quantitative assessments of the similarities as such, and character weighting is minimized because of the lack of any background knowledge underlying it. The main task is to produce such a pattern of similarity relations among phenons in which the differences within each of them are minimized and the differences between them are maximized with this pattern being transformed subsequently into a phenetic classification.

Numerical phyletics [83–87] develops quantitative methods for implementing the phylogenetic program (see Section 3.7 on it). Its methods are designed to facilitate the reconstruction of phylogenetic relationships. Accordingly, similarity is considered as an indicator of kinship, and characters are weighed to select the most reliable indirect evidence of kinship. Construction of phylogenetic trees is narrowed down to a formal graph theory without any biological considerations [88]. An ultimate goal of numerical phyletics is to develop a tree-like structure that can be interpreted as a network of kinship relationships among supposedly monophyletic groups, and, thus, is capable of being transformed into a phylogenetic classification.

Theoreticians of the numerical program, based on their scientific and philosophical preferences, see its undoubted advantages in objectivity, formalization, repeatability, and exactness of the classification techniques. However, the latter hold initially one fundamental limitation provided by axiomatic justification of pure mathematical methods. The point is that any formal axiomatics contains a certain element of subjectivity [89,90], so an objectivity attributed to both the methods developed on its basis and the results of their application, considered philosophically, is nothing more than a myth, although very widespread. Actually, it is eligible to discuss whether a method is true or false in the logical sense only with respect to the axiomatics underlying it, but not with respect to a particular Umwelt analyzed with it. Thus, it is hardly possible at all to say if a classification obtained by a pure formal method is true or false as a cognitive model (in the sense of Wartofsky, see Reference [23]) of this Umwelt, which makes no sense in any consideration whether such a classification is objective or not [4]. The only point that can be discussed rightfully is intersubjectivity [53,91], which means that different researchers, solving the same standard problem with the same standard method, get the same standard result (notorious repeatability). As for the exactness of the axiomatically substantiated methods and the results of their application, it is determined only within the framework of the formalizations embedded in the initial axiomatics, and not necessarily so in others [89,90].

Opponents of this program consider its main idea—primacy of the formal method over biological content—flawed. It reduces biologically meaningful tasks to purely technical ones and, thus, from a metaphysical perspective, "puts the cart in front of the horse". As a result, the problem of instrumentalism arises, which means that it is the method as such that dictates how an Umwelt should be analyzed, so properties of the former indirectly shape properties of the latter [11,92]. For instance, application of a hierarchical classificatory algorithm necessarily provides a hierarchically arranged classification, even if the respective real diversity pattern can be non-hierarchical. Extreme formalizations implied by this idea are considered its main drawback from a biologically meaningful standpoint, as they presume inevitably undesirable reductions (see above).

One of the important methodological problems of the numerical program results from a variety of quantitative methods not reducible to either the most general or the most correct [25]. In such a pluralistic situation, the same question inevitably arises, as in the case of the logical taxonomic subprogram (see previous section). Now it is about selection of a particular method among several available. Two general solutions are possible here. On the one hand, the above principle of onto-epistemic correspondence can serve as a basis for such a selection. A method is preferable if it is more effective with respect to the biological content of the task being solved. On the other hand, the choice of a method should be justified "technologically". The better it is formulated within a well-founded axiomatics, the more preferable is it. The first approach is attractive from the point of view of biology, but it contradicts the ideology of the numerical program. The second approach, advocated by proponents of the "mathematical taxonomy" [93], as noted above, subordinates the solution of biologically sound tasks to the authority of the formal method and, thus, brings forth the problem of instrumentalism.

If philosophical questions are left aside, then undoubted practical advantages of the numerical program come to the fore. One of them is that numerical methods make possible comparative analyses of large data arrays. This is especially true for the numerical phyletics operating with many thousands of unit characters (nucleotide base pairs). A possibility of quantitative comparisons of different classifications by their characteristics, as well as elaboration of the consensus classifications for those derived from incompatible datasets, are also among the practical advantages of this program. At last, computer experimentations with virtual models make it possible to simulate macroscopic phenomena that are fundamentally unobservable and not amenable to direct experiments, such as the structure of biodiversity, global phylogeny, etc.

### 3.4. The Typological Program

The typological way of perceiving and representing the qualitative structure of the world is among the most basic aspects of cognitive activity [94]. It begins with personally perceiving and thinking of

nature with gestalts, i.e., integrated images expressing essential features of its various manifestations (aspects, fragments, etc.). The results of such an intuitive perception are then transferred to nature itself. From this, an antique conception of archetype as an initial ideal form ("matrix"), giving rise to a diversity of all real forms, emerged. It also occurred in the general idea of the prototype underlying the natural philosophical concept of the Scala Naturae, which had a significant impact on the formation of the worldview among many naturalists of the 18th and 19th centuries [95].

Typological views are usually derived by authorities from the essentialist ones, but this is hardly true. Aristotle's understanding of essence (ousia), which forms the basis of essentialism in its widespread understanding (ascending basically to Popper), is functional [96,97]. It is this capacity that inspired many taxonomists from scholastics (such as Cesalpino and Tournefort) to early post-scholastics (such as Jussieu and Strickland). Contrary to this, the typology proper, as it appeared in the works of French and German anatomists at the end of the 18th and the beginning of the 19th centuries, was based on an idea of prototypes or archetypes understood structurally, as determined by spatial interrelations of the body parts of organisms [98,99]. Therefore, the typological program was undoubtedly an original product of the early post-scholastic development of systematics with one of its predecessors having been most likely an Aristotelian I. Jung with his conception of the geometric construction of plants (see Reference [11] for details). In the first half of the 19th century, this program dominated, especially in systematic zoology.

The central element of the typology in general, and of the typological theory in systematics in particular, is the type concept. The latter is very multifold and combines many different meanings reflected in a rather rich terminology, so such a kind of typological pluralism is to be taken into consideration when typology is discussed in general. Therefore, before exposing this program, it is appropriate to consider the main attributions of this concept.

In its broadest natural philosophical sense, a type is likened to a natural Law of Nature. According to the Swiss biologist A. Naef, "organisms relate to the type in the same way that events relate to the law they manifest" [100]. From this point of view, both a physical or a chemical law and a type thus understood are equally fundamental attributes of nature—though not directly observable and rather conceivable, but, nevertheless, completely material as metaphysical natural phenomena.

In a more concrete and yet quite natural philosophical understanding, a type is thought of as a kind of generalized structural characteristic of an organism, considered in a generalized (idealized) form. Such type can be expressed as a general body plan (Cuvier) or as a developmental plan (von Baer) or as a metamorphosis of parts of organismal archetype (Goethe). Thus understood, such a "natural philosophical (arche)type" plays a key role in the initial development of the concept of structural homology (R. Owen) without which no systematic (or any comparative) research is possible [33,34].

More empirically understood, a type corresponds to a combination of properties that are characteristic (typical) for a certain group of organisms (or eventually, any other objects). Such a group can be distinguished by a researcher on the basis of various reasons. It can be either a taxon in its proper typological understanding, or a monophyletic group, or a life form, or even just a phenon, etc. Thus, such an empirical type is largely viewpoint-dependent: it is the researcher who decides, guided by a particular concept, which kinds of groups are to be recognized and which properties are to be considered as constituting their types [101].

It is the natural philosophical concept of the archetype that is central to the typological research program in biological systematics. This program realizes a typological theory, according to which the Systema Naturae is a hierarchically ordered diversity of the (arche)types of various levels of generality, with the most fundamental taking the highest position in this hierarchy. This conception underlies a general idea of natural classification as the one most adequately representing the presumed hierarchical structure of the diversity of the archetypes. Respectively, taxa are recognized following the general principle of the unity of type. Each typological taxon is defined by an archetype of a certain level of generality most fully expressed in the organisms belonging to that taxon. For this, the characters used

to recognize taxa are weighed and ranked in a special way: the most significant are those that allow us to characterize most completely the archetypes and their subordinations.

It is clear from the preceding that, in elaborating particular typological classifications, the analysis of archetypes is primary with respect to the classification of organisms [11,61,101]. This means that the hierarchy of archetypes is revealed first. Then the weighting/ranking of the characters attributed to them is carried out, and the typological taxa are diagnosed by these characters. In order to proceed properly from the hierarchy of archetypes through the analysis of the characters to the hierarchy of taxa, the principle of ranks coordination is introduced. Characters attributed to the archetypes of certain levels of generality are used to define the taxa of the same levels. This principle ascends to the methodologies of the French naturalists A.-L. de Jussieu and G. Cuvier. Like taxa of the onto-rational taxonomy (see Section 3.2), typological taxa are thought of quite realistically as the natural kinds. Accordingly, an objective status is also presumed for their taxonomic ranks [101].

The typological program most fully implements the general ideas of the natural philosophy-based typological theory (or theories) at the macro-systematic level, where differences in archetypes are most evident. At the lower (generic and especially species) levels, its capabilities are significantly lower, since the differences between organisms at these levels do not usually affect the body plans.

The natural philosophical typology has been developing from the very beginning in three main versions, which are known as stationary, epigenetic, and dynamic. They were added subsequently with several other versions, adapting the original ones to an evolutionary idea [11]. All of them might be treated as particular taxonomic theories of the typological kind.

For stationary typology, going back to the ideas of the French naturalists F. Vicq d'Azyr, G. Cuvier, and É. Geoffroy Saint-Hilaire, the central is an idea of a structural general plan (bauplan), determined by the spatial (geometric) relations of its components in adult organisms. The overall organismal diversity is structured by detailing these plans from the most to the least general. Accordingly, the typological unity of taxa appears as a unity of the structural plans of the organisms belonging to them.

In epigenetic typology (i.e., epitypology) originating from ideas of the German naturalist K. von Baer, the general plan is considered as coming through ontogenesis. The epigenetic type is mainly a type of individual development of an organismal structural plan. In modern terms, such a developmental type can be interpreted as an ontogenetic pattern. The general structure of the diversity of these patterns can be represented by a branching tree, which was suggested to call "Baerian" [102]. The general idea of the modern ontogenetic systematics is most close to this version of typology. In it, the diversity of taxa is analyzed from the standpoint of the diversity of the ontogenetic patterns of the respective organisms and the taxa are diagnosed by specificity of the patterns characteristic to their organisms [102–106].

Dynamic typology goes back to the ideas of the German poet and naturalist W. von Goethe. It considers the general organismic construction also in a development, but the latter is understood as an ideal (imaginary) metamorphosis (transformation) of different parts of an imaginary archetype of some superorganism. Therefore, it is sometimes called transformational typology [107]. In accordance with its principal idea, a taxon is characterized by a common pattern of particular transformations of the basic archetype. This typological theory served in the mid-19th century as a basis of the typological concept of homology (Owen). In the 20th century, it enjoyed popularity among some constructional morphologists [100,108–110].

From the second half of the 19th century, and especially in the 20th century, the typological program appeared to be in a "shadow" of the phylogenetic program and was almost completely rejected by the phenetic one. Two main arguments were put forward against it: (1) the type is an ideal construction, to which nothing corresponds in nature, and (2) the type is unchanged, which contradicts the central evolutionary idea of recent biology. To some extent, such a negative attitude is aggravated by the fact that the biochemical and especially molecular genetic attributes most preferred in contemporary systematic studies seem not to be amenable to the classical typological interpretation [111].

However, the traditionally negative attitude towards typology [4–6,8,112] began to be replaced gradually by a more positive one in the second half of the 20th century. Some of its key ideas were supported by the "new essentialism" [113,114], which is acknowledged to be compatible with the evolutionary ideas [115–117], up to a proposal of the phylogenetic typology concept [118]. Very interesting in this respect are evolutionary typological concepts of the dynamic archetype (phylocreod) [119–121] and the phylotype [122–124]. They refer to the stable trajectories of the evolutionary development of both particular morphological structures and integrated ontogenetic patterns. The most significant in its promising perspective seems to be a merging of the classical epitypological and phylogenetic ideas about historical formation of the structural plans of organisms with the most recent ideas on genetic regulation of ontogenesis within the framework of the evo-devo program in systematics (see Section 3.8).

*3.5. The Biomorphic Program*

In the basic structure of biodiversity, two principal components are usually recognized, namely, ecological and phylogenetic ones [125]. The former corresponds to the hierarchy of ecosystems, while the latter corresponds to the hierarchy of monophyletic groups. However, according to a wider concept, there are three such components. To those just indicated, the biomorphic component should be added, which is a hierarchy of biomorphs, or life forms [126]. The first component is studied by ecology and is outside of the scope of systematics. The second is explored by phylogenetics shaping a specific Umwelt for the phylogenetic program in systematics (see Section 3.7). The third component is explored by ecomorphology (in its taxonomic meaning, see below), or biomorphics. It is usually taken out of the limits of proper systematics, but recently was suggested to be included in the scope of the latter [11,127].

The need to develop natural classifications of the basic life forms rather than artificial diagnostic keys of scholastic systematics (like that of Linnaeus) was declared by the German naturalist A. von Humboldt at the beginning of the 19th century. This might have become one of the most important ideas in early post-scholastic systematics, but Humboldt's idea left no evident traces in the then prevalent taxonomic theories [11]. Sufficiently developed classifications of the life forms in botany and zoology began to appear in the late 19th and early 20th centuries [128,129]. Their main purpose was to reflect the diversity of the complexes of morphophysiological adaptations of living beings as elements of the ecosystems. By the end of the 20th century, a movement in this direction led to the emergence of a fairly developed theory, which was proposed to designate ecomorphology [130,131].

However, the latter term has two meanings. One of them is connected with an ecological interpretation of organismal morphology with its main task being an analysis of morphological adaptations [132]. Another is associated with the classification of organisms according to their ecomorphological (or biomorphological) similarities (references above). Taking this ambiguity into account, an introduction of the above term biomorphics to designate only the taxonomic aspect of ecomorphology and to consider it as one of the research programs within biological systematics seems justified.

The main task of the biomorphic program is to recognize the biomorphs of different levels of generality and to develop biomorphic classifications on this basis. Biomorphs are understood as groups of organisms distinguished by their bio-morphological (eco-morpho-physiological) specificity. Thus, biomorphically defined taxa unite organisms similar in their morpho-physiological features, which ensures fulfillment of similar ecosystem functions. Thus, the definition of a biomorph includes neither indication of kinship nor the time and place of the origin of organisms, nor their phenetic similarity as such.

Such interpretation of biomorphs provides the research program under consideration with a particular specificity that is not observed in other branches of biological systematics. As a matter of fact, different organisms of the same species, and even different stages of development of the same organism, can belong to different biomorphs if they differ significantly in their bio-morphological

characteristics. For instance, biomorphologically different can be conspecific plants depending on their growth conditions, or insect larvae and images playing significantly different roles in the ecosystems. Thus, not the total organisms, but rather such elementary units ("bricks") of biomorphic diversity are united in the taxa in respective biomorphic classifications [133].

The elaboration of biomorphic classifications begins with character weighting and ranking. The most significant are those describing the most important adaptive features of organisms as elements of the ecosystems. Their ranks are determined by the levels of generality of the corresponding adaptations. In the analysis of characters, any distinction between homologies and analogies does not matter—the general morpho-physiological organization (an "adaptive syndrome") of living beings is considered instead. Based on the characters thus weighed and ranked, the entire classification of biomorphs is built up in which the hierarchy is determined by the hierarchy of respective morpho-physiological organizations of various levels of generality [130,131]. Thus, the classification algorithm of biomorphics is deductive: a common basis for dividing the world of living organisms (for example, a type of metabolism) is initially identified, and then the entire classification is construed from top to bottom following a hierarchy of the syndromes detailing the chosen basis successively.

Thus, the biomorphic program is very similar, by its general classificatory algorithm, to the typological program (see Section 3.4 on the latter). Their fundamental likeness is in considering a timeless aspect of diversity of organisms and classifying them based on a prior character weighing and ranking. A significant difference is that typological classifications are based mainly on homologies, while biomorphics takes into consideration the entire adaptive syndrome of features. At the same time, special emphasis on the functional significance of characters places biomorphics close to the Aristotelian essentialism (ousiology), in which particular attention is paid to the functional destiny of the organisms' parts (see Section 3.4).

From an ontological point of view, this research program takes a very realistic stance toward biomorphs. This is substantiated by reference with the processes in natural ecosystems shaping the entire structure of biomorphic diversity. With this, it is often assumed that there is only one single system of biomorphs because there is, supposedly, only one global functional structure of the entire biota [130,131]. This idea ascends evidently to the Humboldtian monistic natural philosophy. However, according to another point of view, it may make sense to develop different biomorphic classifications in which the same organisms can be allocated to different taxa [134–136]. All this seems to be similar, to a degree, to distinguishing between "general purpose" and "special purpose" classifications in the phenetic program (see Section 3.1) and means a certain balance between taxonomic monism and pluralism within biomorphics.

The program under consideration, by elaborating its biomorphic classifications, is of importance for that division of ecology dealing with analysis of the structure of ecosystems. The biomorphs recognized in such classifications relate, in a certain way, to classifications of the syntaxa considered, according to one of the synecological conceptions, as fundamental units of that structure [137]. On the other hand, properly construed biomorphic classifications provide very significant information for the analyses of the interrelation between structures of phylogenetic and biomorphic aspects of biodiversity, as they are shaped in the course of biological evolution.

*3.6. The Biosystematic Program*

The term biosystematics has two meanings. On the one hand, it is sometimes used to refer to the entirety of biological systematics: it is simply a contraction of these two words. On the other hand, this is one of the research programs in systematics dealing predominantly with the study of species and intraspecific diversity. In this paper, this term is used in the second sense.

Biosystematics thus understood appeared to be one of two main programs, along with phylogenetics (see the next section), implementing evolutionary ideas in biological systematics [138]. Its title biosystematics was meant to emphasize its main concern with the natural living populations, and not with the dead museum specimens, and it became likely the first to have been officially called

evolutionary systematics [139]. Its principal conceptual source was the classificatory Darwinism of the second half of the 19th century, which was emphasized by calling it Darwinian systematics [140]. Accordingly, the beginners of this program declared persistently that the main lower taxonomic units are not "Linnaean species" but geographic races, which are the only natural biological entities deserving exploration and classification [138,140–144]. The term population systematics directly indicated the level in the taxonomic hierarchy it deals with [144]. At last, to stress a novelty of this program against the orthodox one, it was termed the "new systematics" [145–147].

Emergence of this taxonomic theory and program appeared to be, along with phenetics (see Section 3.1), a specific response of biological systematics to the challenges of the positivist philosophy of science. No less (if no more) important role in its shaping played an active formation of the new contents of biology at the beginning of the 20th century with its interest in ecology, physiology, genetics, and in its efforts to explain everything by evolutionary mechanisms acting at the population level. Biosystematics absorbed new ideas and facts, considering them from a standpoint of the evolution of populations. Due to this, it played an important role in the formation of an evolutionary concept called the "Modern Synthesis" in the 1930s–1940s [147].

According to the specific understanding of its tasks, biosystematics (together with phenetics) abandoned the general idea of the global natural system for the simple reason that supra-specific systematic categories were excluded from its particular Umwelt. It is mainly engaged in elucidating the ecological nature and genetic mechanisms of both the dynamics and the stability of intraspecific categories and units called gene-ecological by the Swedish biologist G. Turreson [148]. It was emphasized that these biosystematic units and their classifications should not necessarily coincide with those of "orthodox" systematics, since they were distinguished on different grounds [149,150]. It was also proposed, in addition to the classification of those units, to fix continuous trends ("clines") of geographic and ecotypic variability of particular characters over the entire ranges of widely distributed species [151,152].

Biosystematic studies focused on comparative analysis of data using all available categories of data to discriminate intraspecific gene-ecological units, thus realizing the phenetic idea. The only significant difference between phenetics and biosystematics, from a taxonomic perspective, is that the former uses only proper traits of organisms (see Section 3.1), while the latter pays attention to their ecological characteristics. This circumstance has predictably caused an extensive employment of numerical methods by biosystematics. In addition, within the framework of this program, a special area of research has been formed, namely, experimental systematics [153,154], which may be treated as a kind of response of systematics to the physicalist challenges, alongside with the above numerical program. It is based on an idea that all judgments about the differentiation of closely related species and intraspecific units should be subject to the experimental verifications under natural and/or laboratory conditions.

Biosystematic research, from the very beginning to the present, have been most popular in botany [154–160]. In particular, one of its leaders, the Soviet biologist A. Takhtadzhyan [157], defined it as "a branch of botany studying the taxonomic and population structure of species, its morphological, geographical, ecological, and genetic differentiation, origin, and evolution". In zoology, biosystematics (mostly under the name "new systematics") was initially promoted by E. Mayr [144], but, later, interest in it almost disappeared [161].

Recent phylogeography, dealing with reconstructions of the microphylogenies of widely distributed species [162–164], may have certain concern for the biosystematics issues. However, it restricts itself by the numerical phyletic methods and does not take into consideration other types of data (morphological, ecological, etc.). Therefore, its results play but an auxiliary role in complex biosystematic studies.

On the opposite side, the recently developed idea of the integrative systematics, as opposed to the total molecularization of research at the species level, can actually be considered a certain revival of the biosystematic theme in zoology. Its main idea is that the delimitation of species units by molecular

genetic characters is only an initial stage in the study of species diversity in which the bulk should involve analysis of all available characters, which allows us to consider various aspects of species' natural history [165–168].

*3.7. The Phylogenetic Program*

This program is another version, along with biosystematics, of the implementation of evolutionary ideas in biological systematics, in this case at macro-evolutionary levels. The first attempt to initiate it was the "Philosophy of Zoology" by the French biologist Lamarck at the beginning of the 19th century. It was actually focused on macro-evolution but appeared to be premature. The second attempt was the "General Morphology of Organisms" by the German biologist E. Haeckel in the second half of the 19th century, which turned out to be much more successful. One of its principal outputs became systematic phylogeny (as Haeckel himself called it), associated with the historical interpretation of macro-taxa and their characters. It is now commonly known as phylogenetic systematics.

The main parameters of the Umwelt shaping the ontological basis of the phylogenetic theory and program can be briefly summarized and formalized as follows [26,27,169,170]. Phylogenetics is based on an assumption (quasi-axiom) that the ordered diversity of organisms is a result of the global long-term phylogenetic process encompassing biota as a whole. This process involves the origin of some groups (descendants) from others (ancestors), and the emergence of new groups being accompanied by the emergence of their specific properties (Darwin's formula descent with modification). It encompasses divergent (cladogenesis) and directed (anagenesis) components. Divergent evolution leads to an irreversible decrease in both kinship and similarity, while anagenetic evolution can lead to a secondary decrease in the similarity in some structures (convergence). Attributes of a newly emerging group of organisms are inherited from its closest common ancestor by the latter's descendants in both conserved and modified forms, and this makes them more similar to each other than to members of other groups (quasi-axiom of inherited similarity). Phylogeny thus understood produces a phylogenetic pattern defined as a hierarchy of monophyletic groups of different levels of generality interconnected by kinship (phylogenetic) relationships. It is evident that both the entire phylogenetic pattern and monophyletic groups within it are treated realistically.

From these basic assumptions, it is deduced that the natural classification should be phylogenetic. This means that any particular classification should reflect the structure of the respective phylogenetic pattern as completely as possible. This general idea is implemented by the principle of monophyly: a group should be characterized primarily by unity of origin, i.e., should include descendants of a single ancestral form. This principle is crucial for the entire phylogenetic program: only the monophyletic group (phylon), characterized by such a unity of origin, is thought to be natural and can be recognized as a taxon in phylogenetic classification. On the contrary, any group that does not meet this criterion is treated as polyphyletic and considered artificial in most schools of phylogenetic taxonomy. Accordingly, in elaborating a phylogenetic classification, the most significant characters are those that allow us to recognize monophyletic groups.

The main contribution of the phylogenetic program for the development of other branches of biology is that phylogenetic reconstructions provide a sufficiently reliable basis for the historical interpretations of similarities and differences between organisms by any kind of trait. One of the instruments of such an interpretation is the detection of the "phylogenetic signal" in the overall pattern of similarity relations, which means a measure of similarity of organisms due to their kinship relationships rather than to ecological causes [171]. Besides, phylogenetic reconstructions play a key role in the historical biogeography.

This program has been dominating in biological systematics since the mid-19th century. It was represented first by what can be reasonably termed classical (Haeckelian) phylogenetics. Two other main versions (subprograms) were added to it in the mid-20th century, namely, evolutionary taxonomy and cladistics. These subprograms basically differ in their treatments of the phylogenetic process (less or more reductionist), the relations between the phylogenetic pattern and the phylogenetic classification

(less or more strict), as well as the methods of elaborating the latter (selection of characters, assessment of similarity, ranking taxa, etc.). Another important difference between them is determined by two particular interpretations of the principle of monophyly, which can be treated as either narrow or broad. In the first case (holophyly), a group is considered monophyletic if it includes all descendants of the same ancestor, with the latter being treated obligatory as a species. In the second case (paraphyly), a monophyletic group includes only part of the descendants of the same ancestor, which may be a supra-specific group. The groups defined according to these two versions of monophyly are termed holo- and paraphyletic, respectively. By all of these features, classical (Haeckelian) phylogenetics and evolutionary taxonomy are close to each other, while cladistics is the most specific.

The classical phylogenetic subprogram equally takes into account both cladogenetic and anagenetic components of phylogeny, though it does not place a particular emphasis on the adaptive nature of evolution. Evolutionary changes are considered mainly as transformations of the structural plans of organisms, according to which their groups are interpreted as various implementations of such plans. Thus, of great importance are the reconstructions of the ancestral body plans, which makes the Haeckelian approach a phylogenetic interpretation of structural typology [100] (see Section 3.4 on the latter). The geological time of existence of the groups is quite significant, as it allows us to treat some earlier organisms as potential ancestors of some more recent ones. The phylogenetic tree in its classical interpretation has a rather complicated configuration: it is "tied" to the geochronological scale and shows a sequence of divergence, time of existence, and dynamics of diversity of the monophyletic groups, as well as successive stages of transformations within respective prevailing anagenetic trends (such as "arthropodization", "mammalization", "angiospermization", etc.).

An emphasis on body plans implies that the monophyletic groups are characterized by commonality of both conservative and innovative characters, some of which can be acquired as a result of parallel evolution. Accordingly, monophyly is understood as "broad", so both holophyletic and paraphyletic groups are equally significant in the phylogenetic classifications of this kind. The main argument in favor of the reality of paraphyletic groups is that they do not lose their morphobiological specificity after cleavage of their "side branches" by developing their own novel specializations [172–177]. Examples are bryophytes and vascular plants, actinopterygians and tetrapods, archosaurian reptiles and birds, artiodactyls and cetaceans, etc.

The correspondence between the phylogenetic tree and a classification based on it is admitted to be soft. It is sufficient that the latter should be compatible with the branching structure of the tree and should not contain evident polyphyletic taxa (such as homoiotherms). Accordingly, the tree being converted into a classification can be cut in different fragments both vertically and horizontally to adequately reflect both the kinship relations and anagenetic specificity of the monophyletic groups. Therefore, generally speaking, the same phylogenetic tree can be equally represented by several phylogenetic classifications that have some different details. The hierarchy of phylogenetic classification in its classical interpretation is ranked.

The evolutionary taxonomy subprogram was designated by its founding father, the American biologist G. Simpson [178], in order to demarcate it terminologically from evolutionary systematics in its biosystematic interpretation (see previous section). This phylogenetic subprogram resembles the classical one by most of its key presumptions. Its specificity is determined by the great attention paid to the adaptive essence of the evolutionary process, with the concept of the adaptive zone playing an especially important role [179]. This makes evolutionary taxonomy similar, to an extent, to the above biomorphics. The most fundamental demarcation between them is defined by including quasi-axiom of evolution in shaping the former's Umwelt. The adaptive zone is defined as a set of environmental conditions that determine the general type of adaptation of organisms. With this, it is assumed that the morpho-physiological specificity of a group, acquired in the course of its evolution, is a result of a similar reaction of organisms with similar epigenetic organization, inherited from their common ancestor, to similar environmental conditions. According to this evolutionary model, the acquisition of a basic adaptive syndrome of a taxon due to parallel evolution of its members witnesses its

evolutionary unity no less than the inherited features. Thus, taxonomic integrity is defined by three interrelated evolutionary parameters, which include the unity of origin (monophyly in its broad sense), the unity of morphobiological organization (anagenetic grade), and the unity of evolutionary trends (parallelisms) [178,180,181]. In arranging phylogenetic classifications, an auxiliary principle of decisive gap is introduced, according to which the levels of mutual distinctiveness of taxa should be taken into consideration in both their individuation and their ranking.

The cladistic subprogram, in contrast to the two just considered, is based on a drastically simplified representation of phylogeny, which is reduced to cladogenesis, and on a respectively simplified interpretation of both phylogenetic relations and phylogenetically significant similarity. The founding fathers of this version, the German biologists W. Zimmermann and W. Hennig, designated their approach as phylogenetic systematics [26,182–185] and this designation currently dominates [27,169,170,186,187]; sometimes it equates to the entirety of biological systematics [187–189]. However, as will be shown below, its differences from the other phylogenetic subprograms are so significant that the term cladistics proposed by the American biologist E. Mayr [190] is more than justified.

In cladistics, the phylogenetic tree is reduced to a fairly simple cladogram, and monophyly is refined to holophyly. The phylogenetic (more correctly, cladistic) unity was determined initially through a cladistic relationship as follows: two groups, A and B, are closer to each other than to another group, C, if the nearest common ancestor of A and B is more recent than the common ancestor of all three groups. In a later version currently dominating, this relationship is determined by reference not to a hypothetical ancestor, but to some real remote group. Two groups, A and B, constitute a holophyletic group if it is shown that they are sisters relative to a third group, C, external to them (routinely called an outgroup). Thus, the concept of ancestor, and, with it, the geological time scale are paradoxically excluded from an Umwelt of this branch of phylogenetics.

At an operational level, the principle of synapomorphy is introduced to reveal the hierarchy of sister groups, according to which the holophyletic group is determined only by synapomorphies, i.e., by similarities in apomorphic (uniquely derived) characters, while simplesiomorphy (similarity in ancestral and "parallel" characters) is not taken into account. This principle makes cladistic theory very peculiar with respect to its logistics [11,169,191]. In all other classificatory approaches, the two-state (Aristotelian) division logic dominates, in which judgments of "A" and "non-A" types (both presence and absence of characters) are equally significant for identifying taxa. In cladistics, the one-state logic of the Soviet logician Vasil'ev [192,193] actually operates. Only judgments of type "A" (synapomorphies) are significant, while judgments of type "non-A" (non-synapomorphies, i.e., simplesiomorphies) are insignificant for the recognition of holophyletic taxa. In addition, the above principle means that, for delineation of a holophyletic group, only its inner similarity is significant, while its outer differences from other taxa are insignificant. Thus, these two components of general similarity relations—similarity and difference—turn out to be logically asymmetric with respect to their classification function. For this reason, the above principle of the decisive gap is not relevant in cladistics. Lastly, the principle of synapomorphy replaces the typological component of classical phylogenetics with a variant of phenetic combinatorics of characters. A cladon (clade in a pure taxonomic sense) is identified more reliably if it is diagnosed with a larger number of synapomorphies ("the more characters, the better"), which presumes an active use of certain numerical methods.

Cladistic classification is based on a strict correspondence between the hierarchy of sister groups in a cladogram and the hierarchy of taxa in the respective classification. For this, the initial cladogram is cut vertically only, with all horizontal relations being discarded, which provides recognition of cladistically consistent taxa (cladons). This is complemented with the principle of equal ranking of sister groups: the groups descending to the same node (branching point) of a cladogram are treated in their respective classification as the taxa of the same rank. As a result, the ranked hierarchy of cladistic classifications for large diverse groups become very fractional and non-operational. This eventually

leads to a suggestion to abandon fixed ranks from the hierarchy of cladistic classifications and to make them rankless [8,194–196].

The general position of cladistics regarding the ontological status of holophyletic groups is declared realistic [26,27,182–184]. However, as emphasized above (see Section 2.1), the more reduced the Umwelt constituting the ontological basis of a particular taxonomic theory is, the less the portion of objective reality of the original Umgebung it contains. This conclusion is clearly true in the case of cladistics: it is based on quite a reductionist representation of phylogeny, and, therefore, has a poorer ontology in comparison with both classical phylogenetics and evolutionary taxonomy.

In the contemporary phylogenetic studies of extant organisms, an approach called molecular phylogenetics (phylogenomics, genophyletics) takes a leading position. It includes the analysis of DNA or/and RNA nucleotide sequences, assessment of the similarity between organisms by these sequences, and construction of molecular phylogenetic trees based on this similarity. All these procedures employ numerical techniques, which makes numerical phyletics (see Section 3.3) an instrumental part of molecular phylogenetics. The transformation of molecular phylogenetic trees into classifications in practical studies strongly follows the above principles of cladistics. Thus, the molecular phylogenetics, from a taxonomic standpoint, can rightly be considered as part of the cladistic subprogram. Therefore, its taxonomic application is sometimes called genosystematics [11,197,198], even though it might be more correct to call it genocladistics.

It is curious enough that the most recent development of phylogenetic systematics means that the history of this biological discipline makes a kind of circle by returning to that stage when the scholastic genus–species scheme dominated [11,199]. One feature of this return is signified by an idea of rankless cladistic classifications, and another by using molecular genetic data exclusively for inferring these classifications, which revives something like a unified division basis.

According to the original intention of the cladism ideologists, their approach should be common for all living beings. This intention is implemented by the universal "Tree of Life" project [200]. However, as the recent results show, the hope for a universal "cladification" (see Reference [201] for this term) of the living matter is hardly warranted. The reason is that the basal fragment of the phylogenetic tree, shaped by the branching patterns of the prokaryote taxa, is not strictly divergent but, rather net-like [202]. This obstructs the elaboration of strictly "vertical" classifications that cladistics seek to achieve.

Currently, a conviction is gradually spreading among systematic theoreticians that cladistics, especially with its molecular appendage, is too reductionist to adequately reflect the structure of taxonomic diversity, even if the latter is simplified to being treated phylogenetically. This is reflected in the appearance of some publications speculating on possible developments of biological systematics beyond cladistics [199,203–205]. However, currently, the cladistic approach is actively developing at the methodological level and still dominating in practical systematic research.

*3.8. In a Shade of Dominance: The Evo-Devo Program*

Generally speaking, this research program is just beginning to take shape and is poorly noticeable against the currently dominating cladistics based on the analysis of molecular genetic data [199]. Its specificity is in that it focuses on the evolutionarily interpreted variety of ontogenetic patterns of multi-cellular organisms [206,207]. The basis for this is provided by a synthesis of the considered phylogenetics in its rather widened sense, epigenetic typology (see Section 3.4 on it), and the evo-devo concept (a well-known abbreviation for the evolutionary developmental biology). The concepts of dynamic archetype (phylocreod) and phylotype (phylotypic stage) mentioned above (see Section 3.4) fit well into the general context of this taxonomic theory. The first means a stable (canalized) trajectory of the evolutionary development of ontogenetic patterns, and the second refers to those patterns that are initial for particular phylocreods and changes of which lead to switching from one phylocrecode to another mainly due to changes in the composition and function of the regulatory genes.

From a historical perspective, this theory goes back to the classical principle of triple parallelism of the mid-19th century. It links (a) the distribution of the body plans of organisms in the natural system, (b) the sequence of appearance of organisms with various body plans in geochronology, and (c) the successions of ontogenetic stages in the individual development of those organisms. Its fundamental novelty is incorporation of the evo-devo concept that focuses on the historical changes of the genetic mechanisms of regulation of ontogenesis [208–213].

As can be seen, the evo-devo (or phylo-evo-devo) taxonomic theory and respective research program are based on a rather rich biologically meaningful ontology, which distinguishes it positively from reductionist cladistics and molecular phylogenetics. This means another, newer version (along with biomorphics and evolutionary taxonomy) of the most recent biologization of the systematics. At the same time, as far as phylogeny is considered one of the cornerstones of this program, it is possible to consider the latter as another branch of the phylogenetic program in its widest sense.

By focusing on the evolution of ontogenetic patterns and the epigenetic mechanisms ensuring their historical stability and dynamics, this research program brings its own version of representing historical patterns of biodiversity and respective classifications. The former can be represented by a phylo-ontogenetic tree, which is actually a phylogenetically interpreted "Baerian tree" (see Section 3.4 on the latter). This tree is transformed into a corresponding evo-devo classification in the same manner as the phylogenetic one with its ranking scale being derived from a sequence of appearances of respective ontogenetic patterns in the evolution of multicellular organisms. The main characteristics of an evo-devo taxon becomes its specific ontogenetic pattern as a whole dynamic system, not reducible to any particular developmental stages [103–105,207]. All this provides biological systematics with a rich ontological basis and allows it to get rid of the overload reductionism brought in it by the above "genocladistics". It deprives the molecular factology of its priority status and overcomes a kind of traditional adultocentrism in the consideration of organismal anatomy [212].

It is evident that the evo-devo research program is not universal: its application is limited to the groups of multicellular organisms with sufficiently developed ontogenetic cycles. Accordingly, many protists and apparently all prokaryotes appear to be outside the scope of its competence. However, this circumstance should hardly be considered a serious disadvantage. As emphasized above (see Section 2.2), any research program in systematics—more precisely, each particular taxonomic theory underlying it—is inevitably local with regard to its applications.

From an epistemological viewpoint, the research program under consideration faces a serious problem caused by its rich natural ontology. The latter presumes that the elaboration of evo-devo classifications should be based on a joint exploration of two complexly interacting multifaceted dynamic systems known as phylogeny and ontogeny [106,214,215]. In such a knotty cognitive situation, the so-called NP-completeness problem (see Babbitt [216]) becomes very relevant. This means that the more complicated the initial conditions of a certain research task are, the less likely it becomes to find its most optimal solution. In the case of systematics, this means a low probability to attain a classification most optimally representing a specific multi-faceted Umwelt shaped by the evolutionarily interpreted diversity of ontogenetic patterns [11,169]. Therefore, elaboration of the evo-devo natural classifications turns out to be significantly more troublesome as compared to phenetically or cladistically consistent ones. However, this problem is true for the evolutionary taxonomy (see previous section on it) as well, as it also deals with a very rich natural ontology.

At the moment, classifications realizing the evo-devo taxonomic theory most consistently and, thus, belonging to the program in question are very few [104,208–213,217,218]. The reason is that detailed studies on diversity and evolution of the mechanisms of regulation of ontogenesis in animals and plants on a modern epigenetic basis are just beginning. Therefore, as always occurs with new disciplines, they involve analyses of only a few model organisms. Therefore, it seems premature to consider how actively this research program will be developing, how productive it may turn out to be for systematics, how serious the alterations of taxonomic classifications may be, and which particular alterations will occur. Among the main tasks to be solved by the evo-devo taxonomic theory, to make

the program in question more promising, seem to be the following: (1) elaboration of a calculus for assessment of the relative significance (weight) of the differences in molecular sequences and ontogenetic rearrangements, (2) elaboration of the general ranking scale for the evo-devo classifications of different groups of organisms, and (3) development of an optimal way to combine vertical and horizontal interrelations between groups with different ontogenetic patterns to reflect most adequately both their primitive (ancestral) and derived features.

## 4. Conclusions: How to Handle Taxonomic Pluralism

As emphasized at the very beginning of this article, taxonomic pluralism is supported by the conceptualism-based philosophy of science. Nevertheless, classical taxonomic monism is also very common, and not only in academic circles but also (perhaps even more so) among practitioners. Thus, the problem of their balance is very relevant for the contemporary systematics.

In a softer formulation, this problem can be represented not by confronting these positions but rather by a question as to how limited taxonomic pluralism could and should be [219,220]. Such a standpoint presumes implicitly that there are more or less good or bad taxonomic theories, so this issue actually refers to how to "separate the clean from the unclean" and to eliminate somehow the latter. This important question was considered briefly from a philosophical standpoint at the end of Section 2. In this section, it is the right place to consider it from a more empirical standpoint.

One of the main practical outputs of taxonomic pluralism at a theoretical level (multiplicity of taxonomic theories) is that it produces taxonomic pluralism at an empirical level (multiplicity of classifications for particular groups). The latter means that the same organisms can rightfully be allocated to different taxa in classifications based on different taxonomic theories. However, various users of taxonomic classifications wish to obtain a unified and stable list of taxa and do not intend to puzzle out differences between particular theories and research programs. Thus, practitioners seem to vote uniformly for taxonomic monism by supposing there is actually only one natural pattern of taxonomic diversity reflected by only one natural classification available for a uniform direct use in various applied projects.

The simplest and most straightforward answer is offered by a pragmatic perspective, according to which the main evaluating criterion for a taxonomic theory is the ability to convert it into an operational concept most effective in resolving certain practical tasks. Such a theory deserves development by providing support for the respective taxonomic community elaborating it, while others are doomed to elimination because of the restricted resources for systematic studies. Quite a demonstrative case in this respect is the recent short but hot discussion of the instability (plurality) of species classifications to be used for the conservation purposes [221–223]. It illustrates how such a "scientific social Darwinism" may turn the disagreement of scientific ideas into an administrative struggle for the limited resources, which, in most recent times, is promoted indirectly by the system of grant support for scientific activity [224].

Another approach, of an epistemological kind, appeals to the scientific consistency of a taxonomic theory that has to correspond to certain criteria, which allows it to distinguish scientific knowledge from others (religious, commonplace, etc.). In particular, it is presumed that such a theory should make it possible to elaborate testable scientific hypotheses about the structure of taxonomic diversity [219]. The problem in this case is that such criteria vary with evolution of the philosophy of science, so the theories consistent from one standpoint may appear to be inconsistent from another. Several decades ago, numerical phenetics pretended to be both the most scientifically consistent and the most effective, and struggled against phylogenetics [5]—and where is this theory now?

Lastly, it is possible to consider this question from a ontological (metaphysical) standpoint, according to which the best taxonomic theories are those that are substantiated by referencing the natural causes structuring biota, and, thus, are biologically meaningful enough (see Section 2.1). From such a perspective, preference should be given to the theories with well-developed natural ontologies modeling multifaceted taxonomic diversity as completely as possible. From this standpoint, any

episto-rational theories (see Section 3.2 on these) are least relevant, whereas, among ontology-based theories, those developed by evolutionary taxonomy and evo-devo research programs seem to be more consistent and more effective as compared to, say, more reductionist cladistics (especially in its genosystematic version).

From a scientific perspective, any culling of some research programs in favor of others, by ascribing a privileged status to the latter, contradicts the fundamental principle of the freedom of scientific activity. From a metaphysical standpoint, the only serious limitation of a rampant pluralism of taxonomic theories seems to be imposed by the very most important task of biological systematics to produce biologically sound classifications representing most comprehensively multifaceted diversity of living matter. Theories that provide such a possibility are considered good while those that do not are considered bad. However, it should be borne in mind that such categorization of taxonomic theories should not be taken as universal; instead, they may appear to be either good or bad locally in different cognitive situations. This is because the structure of the diversity in different groups of organisms can be shaped by different causal factors, so the most pertinent (locally good) to them may be different partial taxonomic theories referring to different Umwelts. At any rate, the optimal theories seem to be those that (a) would embody the advances and diminish shortcomings of all three above ideas concerning the limitation of taxonomic pluralism, and (b) would be flexible enough to allow to take into account biological specifics of particular groups of organisms.

Facing irreducible multiplicity of the research programs in biological systematics, another more relevant question seems to come to the fore [11,17]. How should different classifications be combined, with each reflecting a particular manifestation of the entire taxonomic diversity, in order to get a whole picture of the latter? As indicated above (see Section 2.1), it hardly seems possible to elaborate a biologically sound integrated or "omnispective" classification. Thus, one of the possible answers to this question may be an appeal to develop something like a combined faceted classification. It would likely allow—for each group of organisms and, eventually, for the entire Tree of Life—to embody different particular classifications based on different taxonomic theories into a single pool. In this connection, one of the most pressing tasks of the general taxonomic theory would become the elaboration of an appropriate meta language with an exhaustive natural (non-formal) ontology uniting those developed under different particular theories.

**Funding:** The Governmental Theme no. AAAA-A16-116021660077-3.3 implemented by the Research Zoological Museum at Lomonosov Moscow State University, supported this contribution.

**Acknowledgments:** I am sincerely grateful to Alessandro Minelli for inviting me to contribute to the special issue on "Renegotiating Disciplinary Fields in the Life Sciences" and for commenting on a draft version of this contribution. Three peer reviewers are deeply thanked for their most useful criticisms, comments, and suggestions.

**Conflicts of Interest:** The author declares no conflict of interest.

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
