# Peer review of "Multiplicity of Research Programs in the Biological Systematics: A Case for Scientific Pluralism"

_philosophies, doi:10.3390/philosophies5020007_

Round 1

Reviewer 1 Report

The strength of the paper lies, in my opinion, in providing an overview of the principal biological systematics’ research programs “regardless of their ‘popularity’”. In doing so, it represents a valuable and useful piece of literature. However, I have some doubts – that the @ might consider addressing – concerning the theoretical, and philosophical, part of the paper. In particular, I think that using Uexküll’s notion of “Umwelt” in order to “unveil” the different ontological commitments and the different epistemologies of the different taxonomic research programs (sec. 2.1) may both (i) “betray” the original meaning of the notion; and (ii) making the @’s approach very subjectivist and constructivist (if embracing a strongly constructivist and subjectivist position about taxonomies is actually the @’s intention, I think it should be explicitly stated).

It seems to me that some concerns in these regards should be addressed. In particular, concerning (i): For Uexküll, the Umwelt is species-specific (i.e. the sum of a species’ perceptive world and operational world; even though Uexküll seems to allow, in the case of our species, a greater intraspecific difference than for other species, meaning different Umwelts for the same species). If the @ agrees with this, then it should be explicitly stated that the Umwelt at issue here is our species’ Umwelt. However, the paper seems to take the Umwelt not in the original Uexküll’s sense, as species-relative, or organism-relative, but rather as relative to a certain taxonomic research program. There is a tension here that, in my opinion, should be addressed: on the one hand, what the @ calls “Umwelt” seems rather to be close to what Quine calls the ontological commitments of a theory, or even to Husserl’s regional ontologies. But, if so, why making reference to Uexküll and not to Quine or Husserl? On the other hand (concerning ii), I think that the @ should address a “Kantian concern”: if Umwelts are to be understood, as Uexküll did, as phenomenal realities, then none of them grasps reality in itself, but only reality as it is perceived (and construed) through our senses and concepts (see for instance chapter 4 of Brentari’s book “Jakob von Uexküll, Springer 2015). And this means buying into a subjectivist and constructionist view. Is this the view that the @ is embracing? A possible way out may be consist in specifying that the Umwelt of our species may coincide with the world as it is seen by science – as it seems to emerge from the very last part of Uexküll “A Foray Into the Worlds of Animals and Humans”.

Some more specific remarks.

SECTION 1

LINE 27: “The principal task of science” is debatable; maybe better: “One of the main tasks; one of the tasks”.

LINE 37: “essential properties”, since in philosophy “essential property” is a technical term, I would not use “essential”, but rather “important” or something similar.

LINE 54: Please, clarify why parametric generalization are the only consistent scientifically.

LINE 56: What is meant by “the non-classical paradigm”?

LINE 57: What is meant by a “finalized concept or theory”?

LINE 69-71: Not all authors agree that biodiversity and biological diversity are synonym; according to some, “biodiversity” refers to a new – more value-laden – concept. (See for instance Takacs’s book The idea of biodiversity. Philosophies of paradise, 1996.) Maybe it would be worth mentioning it?

LINE 96-97: not entirely clear; rephrasing suggested.

LINE 97-99: I think that some references should be provided.

LINE 103: “if naturally individualized”, please clarify what this means

LINE 127: “substantiated”, please clarify what this means

SECTION 2

LINE 137: the label “scientific philosophy, considered in general” is an uncommon expression, please define or replace with “general philosophy of science”.

LINE 138: it seems to me that “systematic philosophy” should be replaced by “philosophy of systematics” (they are entirely different things).

LINE 142-145: the claim that no sufficiently well-substantiated TT is known to exist currently is maybe too bold? Moreover, what “well-substantiated” means, and why being “too formal” is a limit?

LINE 146-150: unclear sentence, please try to rephrase it.

LINE 156-157: the definition of TT seems to imply that there is just one TT, how is this compatible with the advocated taxonomic pluralism?

LINE 172: the sentence is not clear to me since I cannot find any previous reference to Hull.

LINE 188: I’d say “cognitive and perceptual” activity – rather than just “cognitive”.

LINE 190-192: I disagree with this, the Umgebung is not a niche (neither fundamental nor realized), since the Umgebung is not relational (it would be there even if there were no organisms).

LINE 203-210: This sentence is not understandable to me, please, try to clarify.

LINE 211: Please, explain what individualizing an Umwelt means.

LINE 219: that a “particular Umwelt is based on a reduction operation” seems questionable to me. I’d say “selection” rather than reduction (the perceptive and operational marks that makes the Umwelt are a part of the Umgebung).

LINE 231: “cognizing subject” seems a weird expression to me, consider replacing it with “knowing subject” (same for “cognized”, consider replacing it with “known”).

LINE 233 (and 244-249): (see also my general comment at the beginning): it is not clear here to whom the particular Umwelts “belong”. Are they Umwelts of different subjects or different disciplines?

LINE 235-236: This sentence is not understandable to me, please, try to clarify.

LINE 255-262: This paragraph is not understandable to me, please, try to clarify.

[SECTION 2.2.] I have a general concern: This section explains how a TT should/might be built, but it does not build it. If I understand rightly, the “building” part is made in making explicit the different Umwelts of the different taxonomic research programs (in the rest of the paper). If so, I think that this should be clearly stated, for the benefit of the reader. Moreover, I think that more space might be devoted to this task (i.e. making the different Umwelts explicit) in the rest of the paper (i.e. to do the same job that has been done for the phylogenetic program also for the other research programs).

LINES 265-267: An example may be of help for the reader.

LINE 277: Not clear to me what “substantiated” means.

LINE 291-293: This paragraph is not understandable to me, please, try to clarify.

LINE 295: Not clear to me what “possible definitions” are.

LINE 312: Please explain what the principle of onto-epistemic correspondence says.

LINE 332-333: It seems to me that there is a discrepancy here: It has been said before that a TT is made of BOTH axioms and rules of inferences. Here it seems instead that it is made of axioms OR rules of inference.

LINE 348: “certain universal fairly formalized reference rules”: an example would help the reader in understanding what they are.

LINE 352: If respective pTTs are universal in their applications, why they are more than one?

SECTION 3

LINE 365-407: As an introduction to the section, this part presupposes a lot of knowledge by the reader. Would it be possible to simplifying it a little?  I think also that explaining what the requirements of empirical science paradigm are (line 377) and what the challenge of the positivist philosophy id (line 389) would be useful.

LINE 370-371: It seems to me that what the @ is referring to is the essence of a species (rather than an organism), i.e. a property or a set of properties that all and only the members of a class must share in order to belong to that class.

LINE 425: Not clear what “this philosophy” refers to. Is the @ meaning something like “the theoretical/philosophical implications” of the phenetic program?

LINE 426: “to exclude its ‘metaphysical’ (declared ‘unscientific’) content”: an example would be helpful.

LINE 434: “However, the latter is not true”: please, explain what is meant here (maybe a reference can be made to the fact that similarity is somehow subjective and resists formalization cf. Quine “natural kinds”).

LINE 441: Please, explain what “phenons” are (it seems to me that this word is here used in a different sense than Mayr’s).

LINE 465-466: Here the paper is talking about the cladistic and molecular version of the phylogenetic program, but at line 466 “phylogenomics” enters in play, what is the relation between it and the two previous programs?

LINE 469: Please explain, even in few words, what the positivist principle of total evidence says.

LINE 471: Please explain what “the philosophy of science that focuses on modern conceptualism” is.

LINE 473-474: “Is impossible in the natural science”, please, please, say briefly why.

LINE 493: Please, explain what a “deductive substantiation” is.

LINE 506: It does not seem to me that Quine is a realist about NK. Maybe better use as reference Putnam, “The meaning of ‘meaning’”?

LINE 525: What is it “the classical systematics”? which is?  If it is the “scholastic program”, I think it would be better to continue to use that label for the reader not to get confused.

LINE 532-533: Not clear whether the two subprograms are explained later. If not, maybe saying some more words about them? If yes, please, try to make it more explicit.

LINE 535: “not a once” sounds like a weird expression to me (and I do not understand it).

LINE 539: Not clear to me what or who “they” refers to.

LINE 542: “if not to count those dealing with identification key”: please, explain.

LINE 545-546: This sentence is not clear to me, maybe try to rephrase it in a clearer way?

LINE 547: Not clear what “this question” is.

LINE 561: Please, clarify what is meant by “the modern conceptualist standpoint”.

LINE 578-579: Reference missing, please add it.

LINE 600-601: Please explain or rephrase what is meant by “an object to be studied is to be ‘adapted’ to the needs of quantitative methods by means of its total reduction” (in particular, reduction to what?)

LINE 628-629: It seems to me that a method can be more or less effective, but not true or false (I also have some doubts about applying “true” and “false” to classifications (line 631), it seems to me that “objective” (line 632) is more appropriate).

LINE 640-642: “The problem of instrumentalism arises, which means ‘closure’ of the cognitive activity of biologists on the method as such”. Please, explain what this means.

LINE 669: “The typological way of perceiving” seems to imply that there are different ways of perceiving (or, maybe, that our perception is determined by our concepts, is this what the @ means? Cf. also the “Kantian concern” that I highlighted in my initial general comment). This would be a strong philosophical position, which would need some arguments or at least an explicit endorsement / explanation.

LINE 679-680: “Typological views are usually derived by authorities from the essentialist ones, but this is hardly true”. I agree if you think of Aristotle (as the @ rightly writes), but it is not if you think of Plato (as Mayr, for instance, does in his paper on typological thinking).

LINE 704-705: Please explain what partonomy = meronomy is, and what it means that “Such type … is related to partonomy = meronomy”.

LINE 715: Please, define what “ideal type” means in this context.

LINE 727-728: Please explain what “the partonomic (meronomic) analysis” is.

LINE 739: The paper seems to imply that the the species level is a macrosystematic level while it seems to me that it is a microsystematic level, but of course I may be mistaken.

LINE 788-790: Literature on biodiversity components is endless, and this claim is, in my opinion, controversial. If you look, for instance, at the most popular definition of biodiversity (the CBD’s one), the main components of biodiversity are the genetic, specific, and ecosystems diversity.

LINE 792: Maybe a few more words, or an example, on what “biomorphs” are can be useful to the reader.

LINE 802: What is meant by “the real ecosystems”?

LINE 808: What is it the meronomic aspect of the diversity of organisms?

LINE 833 and 835: Please define what the “adaptive syndrome” is, and what is meant by “the hierarchy of respective syndromes”.

LINE 847: “Ontological interpretation” sounds as a weird expression to me; I would suggest thinking of replacing with something like “From an ontological point of view, this research program takes a quite realist stance towards biomorphs…”

LINE 859-861: Please explain/rephrase in a clearer way this sentence, in order to make it understandable also to non-expert readers.

LINE 868: Please, consider replacing “An abbreviation” with “A contraction”.

LINE 886: “New biology”: Please, include a definition and/or a reference.

LINE 809: “Great Synthesis”: Since the standard label is the “Modern Synthesis” (as in the Huxley’s book’s title, rightly mentioned by the @ himself), the choice for the label “Great Synthesis” should be explained, maybe in a footnote.

LINE 895: “intraspecific categories and units called gene-ecological”. Some lines above, the paper says that the focus was on geographical races, then on populations, now on gene-ecological units. This may be confusing.

LINE 907-908: Formulated like this, it seems that experimentalism and physicalism are quite the same thing. Maybe rephrase?

LINE 940: Not clear what “it” refers to; maybe better to repeat the subject.

LINE 944: Why “directed (anagenesis)” (is it “directed” understood in a Lamarckian sense?) rather than “gradual (anagenesis)”?

LINE 965: Please, define “phylogenetic signal”.

LINE 971: Is “reductional” = “reductionist”? Not clear to me what “reductional” mean.

LINE 988-989: “earlier organisms are perceived as ‘potential’ ancestors of more recent ones” is not clear to me (How the reverse would be possible?)

LINE 1034: If I understand rightly, it is claimed that the term “cladistics” seems nearly forgotten, but it does not seem so to me (for instance, looking for it on Scopus, in the last 10y, around 200 articles have been published each year with the term “cladistics” within the title).

LINE 1060: the term “cladon” sounds weird to me (I’m used to klados (if in Greek) or clade (if in English), but maybe “cladon” is a technical term I’m not aware of.

LINE 1075-1076: by “less realistic” is it meant that cladistics has a poorer ontology? If so, it seems to me that “has a poorer ontology” is a more appropriate expression to express the concept.

LINE 1163: “in the festschrifts on evo-devo cited above”: maybe better to replace it with the number of the relevant bibliographical entry.

CONCLUSION

LINE 1718: What is it meant by “the contemporary non-classical philosophy of science”?

LINE 1187: Who “they” refers to? (Moreover: if I understand rightly, the paper claims that taxonomic pluralism is a minority position, but, at least in philosophy of biology, I am not sure this is true, think of Kitcher, Ereshefsky, Dupré, Minelli, Mishler and Brandon).

LINE 1202: What is it “scientific Darwinism”? Maybe it could be rephrased as “scientific social Darwinism”?

LINE 1214-1216: Here the @ might consider mentioning Kitcher 1984 (“Species”) and Ereshefsky 1992 (“Eliminative Pluralism”).

LINE 1223: What is it meant by “artificial culling” exactly?

Finally, notice that several typos still remain in the manuscripts that should be fixed, and English would require to be revised by a native speaker.

Author Response

All references to particular lines correspond to those in the edited text

Reviewer 2 Report

Basically, I think the thesis is sound and the author has obviously done a great job researching the subject and has a good understanding of the literature. My main concern is how it is all presented. The paper is very long and contains a lot of information that, though interesting, does not seem necessary to make the point the author wants to make. In fact, much of the content would be a better fit for a book. I would strongly suggest that you try to focus more on your line of argumentation and clarify why and how the facts you line up support your thesis. The language is not always very good. In particular, you frequently neglect to distinguish between singular and plural and between definite and indefinite form.This has to be attended to. I strongly recommend using a native English speaker or a professional language editor. Most of the text is also written in a rather cumbersome style which makes it a bit hard to follow.

Author Response

(The authors gave the same response as above.)

Reviewer 3 Report

This paper is a very interesting and potentially useful contribution to the literature, but it also has several problems. The paper presents an overview of the various research programs and theories in biological taxonomy and systematics, addressing the main historical ways of thinking (Section 3). The paper also makes a case for pluralism about taxonomic approaches and explores how we can deal with a plurality of approaches (Section 4 and to a lesser extent some parts of Section 1). This is very useful and these parts would be publishable.

A major problem, however, is the way in which the paper is framed. Section 1 is not clear at all, and Section 2 seems very disconnected from Sections 3 and 4. At least, I did not see how these parts were connected and why section 2 was needed at all. I think the author needs to do much more work to develop the necessary connection and parts of the content of the paper (as I explain below). To some extent it seemed to me that section 2 and parts of Section 1 could simply be deleted, and Sections 3 and 4 would still constitute a worthwhile paper.

The beginning of the paper does not do a good job introducing the topic of the paper. Frankly, the first part of the text was confusing with respect to what the paper was about and what it was trying to achieve. The author starts with a narrative on two modes of doing science without making clear why the readers should know this, or what issues the paper is going to address. I didn’t see why lines 1-66 were necessary and I have the feeling that if the paper would just start with line 67 (“Biology is one of the most…”) it would actually start in a much better way.

In general I had a hard time seeing what exactly the author wants to achieve with the paper even later on in the text, after having read the whole introduction. At the end of Section 1 the author writes that the paper aims to give an overview of the various research programs in taxonomy and systematics, and to show why biological taxonomy is pluralistic. Those are worthwhile aims, but: (1) An overview of the sort given in Section 3 is not a new contribution. There are several such overviews in the literature, most prominently Ernst Mayr’s large The Growth of Biological thought. So, what is new about this overview that would make it a publishable contribution? The author should explain this. (2) From the overview in Section 3 alone it does not follow that biological taxonomy is necessarily pluralistic. After all, it largely is a historical overview and many biologists would simply say that most of these old theories are obsolete. So, the author should strengthen the argument for pluralism.

While (1) and (2) are the main problems (which I think would require considerable work to resolve), below I list some slightly smaller problems.

The introductory discussion on taxonomic monism and pluralism is useful and clear, but also a bit superficial. About pluralism the author writes that all taxonomic theories and programs are of equal biological meaning and status – but I don’t think this is correct. Some theories may be much more widely applicable than others, for example, so the author should say more about this matter. Consider the various approaches in phylogenetic systematics: pattern cladistics, process cladistics, evolutionary systematics, and so on (some of which are mentioned in the part on pluralism in Section 1 – some are much more widespread and endorsed than others, so saying that all of these are of equal meaning and status does not correspond to the way phylogeneticists would see it. Also, consider the debate between proponents of typology and proponents of phylogenetics – there are much less defenders of typology and by far most biologists would agree that typology isn’t a good approach to taxonomy, so how could the author say that both are of equal value and status? In general, it is possible to be a pluralist without holding that all theories, programs and schools of thought are equally important. This part of the text should be made more adequate to the actual state of affairs.

At the beginning of section 2 the author writes: “no sufficiently well-substantiated TT is known to exist currently. Moreover, hardly any satisfactory understanding seems to exist among taxonomists of what kind of theory it should or could be.” I found this a strange remark, as a large majority of biologists agree that taxonomy should be based on common descent (as Darwin already asserted in the Origin) and subscribe to some form of phylogenetic theory following Hennig’s book Phylogenetic Systematics. Of course there are very deep differences of opinion on the details of the best theory, but there still is general agreement that some sort of phylogenetic theory should be chosen. This is different from the situation amongst theoretical biologists and philosophers of biology – they see the plurality of theories more than practicing biologists, I think.

I did not understand the role of Section 2.1 (on Uexküll’s concepts of Umwelt and Umgebung) for what the paper is trying to achieve. Is it an additional argument for pluralism? If so, I don’t think it works well, as the author does not show why Uexküll’s concepts would be applicable to science or biological taxonomy in particular, and also does not show how pluralism follows from applying Uexküll’s work. And if it isn’t an argument for pluralism, then what does the author want to argue in this part of the paper? In lines 248-250 and at the very end of the paper (Section 4), the author refers to Uexküll’s concepts in relation to the biological meaningfulness of classifications. But I found this quite confusing, because most biologists do not use Uexküll’s concepts at all. In my view, biological meaningfulness should connect to actual biological practice, but by using Uexküll’s concepts the author moves the paper quite far away from what is actually happening in biology.

Section 2.2 (on quasi-axiomatics) suggests that the discussion of Uexküll’s concepts is necessary to develop the ideas in this section. But I had the feeling that it was completely possible to formulate the key ideas of Section 2.2 (the need for a high-level general taxonomic theory and lower-level particular taxonomic theories, and the inherent fuzziness of central concepts) without using any of Uexküll’s concepts. I did not see what Uexküll’s concepts added to the discussion in this section. In general, I found this section informative and think it is a main contribution of the paper, but also found it insufficiently well-developed. I think the author should develop the ideas in this section more deeply and also more clearly.

The text needs considerable language editing and correction, as it contains grammatical errors and incorrect ways of phrasing. I would suggest that the author ask a (quasi-)native speaker of English to have a look at the text.

Author Response

All references to particular lines correspond to those in the edited text.

Round 2

Reviewer 2 Report

I am fine with the paper the way it looks now. I only have some minor issues with the grammar, and some clarifications are needed as indicated below.

(the numbers refer to line numbers)

23-24: Maybe you should add 'Scientific pluralism' among the keywords.

35: Why 'actually'?

57: the structure

59: some of its key ...

61: 'the multiplicity' or 'a multiplicity'?

64: You say 'most important' but do you actually mean 'most specific'?

73: 'the above taxonomic diversity' - do you mean 'the above mentioned taxonomic diversity'?

77: 'been being' does not make sense. Remove 'being'.

241: a number of philosophical texts

257: What do you mean by 'too formal' and why is this a problem?

268-267: Strange sentence. Try to reformulate.

314: You say 'a cognitive situation the natural sciences deals with.' Do you mean 'the cognitive situation dealt with by natural science'?

323: In the case

326 seems to be by indication of (I guess that's what you mean.)

339 "Occam's razor" not the "Occam's razor".

471: 'most reach' - do you mean 'mostly reached'?

472: 'most depleted' - do you mean 'mostly depleted' or 'the most depleted'? (It makes a big difference).

626: 'including being biologically'

702: 'context' not 'contexts'.

709: 'accents' - do you mean 'focus'?

711 'the order' not 'an order'.

713 'the most'

721: 'the French' not 'a French'.

725: 'the German, not 'a German'. This is a recurrent problem. Since you are not talking about just any German naturalist but a specific one, you need to use 'the' not 'a'.

846-847: 'often almost identified'. I am not sure what you mean here. Do you mean that they are often being conflated?

883: 'supposed' - I suggest 'assumed'?

910-911: in the above sense

935: 'the Swiss' not 'a Swiss'.

939: 'the German' not 'a German'.

948: 'one of Driesh's ideas' (not the).

1020: 'a matter of fact'.

1029: 'taken into consideration'.

1030. Is it not possible that there is a general logic even though there are several logical systems?

1040: pure "logical" - I suppose you mean "pure" logical ?

1081: 'the Italian'

1082: 'the Prussian'

1085: 'the English'

1089: 'the Russian'

1102: 'a lot' - I propose 'several'.

1110: is acceptable

1151: 'to be' - I suggest 'of being'.

1161: in the sense

1175: Singular or plural?

1179: 'to either' - not 'to any one either'.

1185: 'it is' not 'is it'.

1196: 'of this'.

1220: What does 'the worlds of things and ideas' mean?

1229-1230: You seem to contradict yourself here. Are they or are they not?

1247: the Swiss

1329: components in

1333: the German

1342: the German

1395: three such components (not three of).

1403: the German

1521: 'concern with' not 'concern to'.

1542: the dynamics and the stability

1543: the Swedish

1544: 'should not obligatory'. I don't under stand. Do you mean 'should not necessarily'?

1560: 'have been most popular' (not being).

1561: the Soviet

1606: the French

1608: the German

1696-1697: the American

1724: their individuation and their ranking

1732: the American

1738: 'a later version' not ' an afterward version'.

1750: the Soviet

1798: If they are not ranked, is it still really a hierarchy?

1906: 'of a certain'

1918: 'not numerous' - I suggest 'uncommon'

1958: What do you mean by "philosophical" (rather than philosophical with out quotation marks)?

1966: practitioners

2038: 'actual tasks' - do you mean 'pressing tasks'?

Reviewer 3 Report

The author clearly has done work to improve the paper, and with success. The author has responded adequately to my comments and it is clear that the remaining points that have not been addressed mainly are differences of opinion. I still have some reservations as regards the aims of the paper (which still aren't stated clearly) and some of the content (the use of Uexküll's terminology as clarifying metaphors). Also, Section 2 in my view does not decisively show that biological taxonomy must be pluralistic - if the argumentation in Section 2 is correct, then all classifications in every area of science must be pluralist (as in all sciences there are many Umwelts/perspectives). But areas like elementary particle physics and chemistry seem to fare quite well with monistic classifications. So, the argument doesn't seem to force pluralism upon us. In addition, the paper still is quite densely written, which makes for a somewhat difficult read.

In any case, I think the paper is sufficiently interesting to be published. in particular the overview of taxonomic theories is very useful. I disagree with the author on several points (see above), but I suppose those disagreements are of the sort that would provide for fruitful debate in the literature.

The language has improved, but I would urge the author to perform one last round of polishing to make the text more reader-friendly and to correct any remaining (minor) language issues. 
